# Chromogranin A deficiency attenuates tauopathy by altering epinephrine–alpha-adrenergic receptor signaling in PS19 mice

Suborno Jati[1], Daniel Munoz-Mayorga[2,7], Shandy Shahabi[1,7], Kechun Tang[3], Yuren Tao[2], Dennis W. Dickson [4], Irene Litvan[2,5], Gourisankar Ghosh [1] ✉, Sushil K. Mahata [3,6] ✉ & Xu Chen [2]

Metabolic disorders such as insulin resistance and hypertension are potential risk factors for aging and neurodegenerative diseases. These conditions are reversed in Chromogranin A (CgA) knockout (CgA-KO) mice. CgA is known to be associated with protein aggregates in the brains of neurodegenerative diseases including Alzheimer's disease (AD). Here, we investigated the role of CgA in Tau pathogenesis in AD and corticobasal degeneration (CBD). CgA ablation in Tauopathy mice (PS19) (CgA-KO/PS19) reduced pathological Tau aggregation and spreading, extended lifespan, and improved cognitive function. Transcriptomic and metabolite analysis of mouse cortices revealed elevated alpha-1-adrenergic receptors (Adra1) expression and high Epinephrine (EPI) levels in PS19 mice compared to WT mice, mirroring observations in AD and CBD patients. CgA depletion in PS19 mice lowered cortical EPI levels and the expression of *Adra1* back to normal. Treatment of WT hippocampal organotypic slice cultures with EPI or *Adra1* agonist promoted, while an *Adra1* antagonist inhibited Tau hyperphosphorylation and formation of neurofibrillary tangles, which is unaltered upon CgA depletion. These findings demonstrate the involvement of CgA in Tau pathogenesis and highlight the interplay between the EPI-Adra1 signaling pathway and CgA in Tauopathy.

Chromogranin A (CgA), a ~ 49 kDa acidic and secretory proprotein, was discovered in bovine chromaffin granules[1]. CgA is co-stored and co-released with catecholamines from their large dense core vesicles (LDCVs) in adrenal medulla and postganglionic sympathetic axons[2]. While CgA was first detected in chromaffin granules, subsequent studies revealed its ubiquitous presence in LDCVs of endocrine, neuroendocrine and neuronal cells[3,4]. CgA expression is elevated in rodent models of hypertension, both genetic (spontaneously hypertensive rat)[5,6] and acquired (renovascular) hypertension[7]. CgA exerts both extracellular and intracellular functions. While the extracellular functions of CgA include generation of bioactive peptides such as Catestatin (CST: hCgA₃₅₂₋₃₇₂) and pancreastatin (PST: hCgA₂₅₀₋₃₀₁), the intracellular function of CgA includes the initiation and regulation of LDCV biogenesis and sequestration of hormones in neuroendocrine cells[8,9]. *CHGA* gene encodes a unique pro-protein harboring counter-regulatory motif. PST is associated with pro-diabetic[10–13], pro-inflammatory[13,14], and obesogenic[13,14] effects, whereas CST[15] exerts anti-diabetic[16,17], anti-inflammatory[16,18], and antiobesogenic[16,19] effects. Additionally, CST

[1]Department of Chemistry & Biochemistry, University of California San Diego, La Jolla, CA, United States of America. [2]Department of Neurosciences, University of California San Diego, La Jolla, CA, United States of America. [3]Department of Medicine, University of California San Diego, La Jolla, CA, United States of America. [4]Mayo Clinic, Jacksonville, FL, United States of America. [5]Parkinson and Other Movement Disorders Center, University of California San Diego, La Jolla, CA, United States of America. [6]Veterans Affairs San Diego Healthcare System, San Diego, CA, United States of America. [7]These authors contributed equally: Daniel Munoz-Mayorga, Shandy Shahabi. ✉e-mail: gghosh@ucsd.edu; smahata@health.ucsd.edu

exerts anti-adrenergic[15,20,21] and anti-hypertensive[18,22–25] effects. Given the importance of CgA in inflammation, insulin sensitivity and hypertension – and the strong correlation of these metabolic orders with neurodegeneration – we hypothesize that CgA may play a significant role in neurodegeneration.

Several studies have highlighted the spatial distribution of CgA mRNA[26] and protein[27] in hippocampus, particularly in the CA2 and CA3 pyramidal cells. However, the functional implications of this distribution remain unclear. CgA was found to colocalize with proteinopathic aggregates and dystrophic neurites in Alzheimer's disease (AD) brains[28–30]. Notably, CgA is present approximately 30% of amyloid plaques, which are surrounded by activated microglia[31]. CgA has been shown to induce the release of Tumor Necrosis Factor alpha (TNFα) from cultured microglia/neurons, as well as neurotoxins from microglia[32], suggesting its potential role in facilitating neuroinflammation and neurodegeneration. A recent study reported increased CgA in the cerebrospinal fluid (CSF) of AD patients, which correlated with phosphorylated Tau (p-Tau) and Tau levels, implicating a potential link between Tauopathy and CgA[33].

Three main types of adrenergic receptors (ADRs) are expressed in the brain such as ADRA1, ADRA2 and ADRB, which play important roles in maintaining normal cognitive functions in humans and in animal models[34]. Animal models and human studies demonstrate that ADRs play a pivotal role in synaptic plasticity, memory and cognitive functions that have been associated with AD[35]. Recently, it has been shown that Adra1s exert short-term and long-term synaptic plasticity in different brain regions such as the hippocampus, neocortex and the prefrontal cortex, regions adversely affected in AD patients[36]. CgA has also been shown to exert its hypertensive[37] and cardioprotective effects via ADRs[38].

Tauopathies, including AD, corticobasal degeneration (CBD), and Pick's disease (PiD), represent a broad class of neurodegenerative diseases characterized by the accumulation of hyperphosphorylated, misfolded Tau species in the brain[39–41]. Tauopathies are closely associated with metabolic dysfunctions, including insulin resistance, and hypertension[42,43], as well as neuroinflammation[44]. Such metabolic dysfunctions are commonly observed in aged wild-type (WT) mice and are reversed in Chromogranin A knockout (CgA-KO) mice[45]. The effects of CgA on metabolism and aging, together with CgA's association with the central nervous system (CNS) and AD prompted us to investigate the role of CgA in the pathophysiology of Tauopathy. We hypothesized that depleting CgA could enhance longevity and ameliorate Tauopathy.

In view of the above, we genetically depleted CgA in PS19 (PS19; human Tau isoform 1N4R with P301S mutation expressed under *Prnp* promoter in Chromosome 3) mice, a well-established model of Tauopathy[46]. Remarkably, these mice exhibited increased longevity, decreased neuropathology, and improved cognitive function. Transcriptomics analysis suggested that the α-adrenergic receptor (Adra) signaling pathway is involved in the role of CgA in PS19 mice. We further performed pharmacological studies with Adra1 agonist and antagonist to validate the involvement of the EPI-Adra1 pathway in an ex vivo model of Tauopathy. Our findings underscore the critical interplay between CgA and the EPI-Adra1 signaling axis in Tau pathogenesis, highlighting a new mechanistic connection between metabolic regulation and neurodegeneration.

## Results

### Elevated CgA levels are associated with Tau tangles in AD, CBD patients and PS19 mice

We measured CgA protein levels in the brain lysates of AD and CBD patients. Western blot (WB) analysis with primary antibody against CgA validated by both immunoblotting and immunohistochemistry (IHC) revealed a significant increase of CgA protein levels in Braak stage VI AD patient frontal cortex and hippocampal lysates compared to Braak stage 0-II (Patient Sample details mentioned in Supplementary Table 1), correlated with higher levels of pathogenic Tau phosphorylation (p-Tau, S202/T205) (Fig. 1A–D, S-Fig. 1A–C). CgA mRNA levels, on the other hand, remained unchanged across Braak stages (S-Fig. 1D). IHC analyses on the hippocampi of AD patients using a CgA-specific antibody showed increased CgA accumulation in Braak VI compared to Braak 0-II (Fig. 1E, F, S-Fig. 1E). Similarly, we found elevated CgA levels in the frontal cortices of CBD patients (Braak III) (Patient Sample details mentioned in Supplementary Table 2) compared to normal controls (Braak 0) (Fig. 1G, H), correlated with increased p-Tau (S202/205) levels (S-Fig. 1F). Consistent with human samples, WB from cortices and IHC from hippocampus showed higher CgA protein levels in the PS19 mice compared to WT littermates (Fig. 1I, J, S-Fig. 1G, H), without any significant difference in CgA mRNA levels (S-Fig. 1I). Co-staining with antibodies against CgA and misfolded Tau species (MC1, conformation specific antibody) in Braak VI hippocampus showed a remarkable colocalization of CgA and aggregated Tau tangles (S-Fig. 1J). As expected, lower CgA immunofluorescence was detected along with no MC1+ staining in Braak I or Braak 0 hippocampus (S-Fig. 1K, L). To directly test the effect of CgA on Tau pathology in neurons, we cultured primary neurons transduced with AAV2-MAPT (P301S) and treated with preformed synthetic Tau fibrils (K18/PL)[47,48]. Treatment with CgA (5 ng/ml) led to marked accumulation of misfolded Tau species detected by MC1 antibody, confirming CgA-mediated augmentation of Tauopathy (Fig. 1K, L). To determine how CgA depletion per se affects Tau phosphorylation and neurofibrillary tangle (NFT) formation, we generated hippocampal organotypic slice cultures (OTSC) from WT and CgA-KO pups (postnatal 7–8 days) followed by AAV-mediated transduction of TauP301S and seeding with synthetic Tau fibrils (K18/PL)[49] (Fig. 1M). Compared to WT slices, we found significant reduction in p-Tau and misfolded Tau species in CgA-KO slices (Fig. 1M, N, S-Fig. 1M–O). These observations indicate that CgA induces pathological Tau burden, whereas its depletion reduces Tau pathology.

### Genetic deletion of CgA reduces neurodegeneration in Tauopathy mice

To systematically elucidate the role of CgA in Tau-mediated neurodegeneration, we bred PS19 mice with CgA-KO[22] mice to generate CgA-KO/PS19 mice. At nine months of age, Nissl staining revealed clear brain atrophy with reduced hippocampal volume in PS19 mice, consistent with the previous report[46]. In contrast, CgA-KO/PS19 mice exhibited minimal brain atrophy (Fig. 2A, B). Representative hippocampal images for each genotype are provided in S-Fig. 2. Morphometric analysis on Nissl-stained brain slices revealed significantly decreased DG, CA1 and CA3 thickness (S-Fig. 3A–C) in PS19 mice compared to CgA-KO/PS19 and WT mice. WB in the cortex lysate validated the depletion of CgA in CgA-KO/PS19 mice (Fig. 2C), where drastically reduced p-Tau (S202/T205, AT8) and p-Tau (S396/S404, PHF1) was found compared to PS19 mice (Fig. 2C, S-Fig. 3D, E). Again, CgA levels were increased in PS19 cortex compared to WT cortex (Fig. 2C). Meanwhile, the levels of postsynaptic density protein 95 (PSD95) were significantly reduced in PS19 cortex lysates compared to WT, consistent with synaptic loss in PS19 mice, which was rescued in CgA-KO/PS19 (Fig. 2C; S-Fig. 3F)[46]. WB also revealed decreased pTau in hippocampus of CgA-KO/PS19 mice (S-Fig. 3G–I). Like in cortex, WB in hippocampus also showed decreased PSD95 in PS19 mice, which was rescued in CgA-KO/PS19 mice (S-Fig. 3G, J). Again, IHC analysis of hippocampus revealed decreased p-Tau in CgA-KO/PS19 compared of PS19 (S-Fig. 3K, L) mice. Furthermore, IHC with the MC1 antibody revealed significantly more misfolded Tau species in the hippocampus (DG and CA3 regions) of nine-month-old PS19 mice compared to CgA-KO/PS19 mice (Fig. 2D–F). These results suggest that CgA promotes Tau phosphorylation and NFT formation.

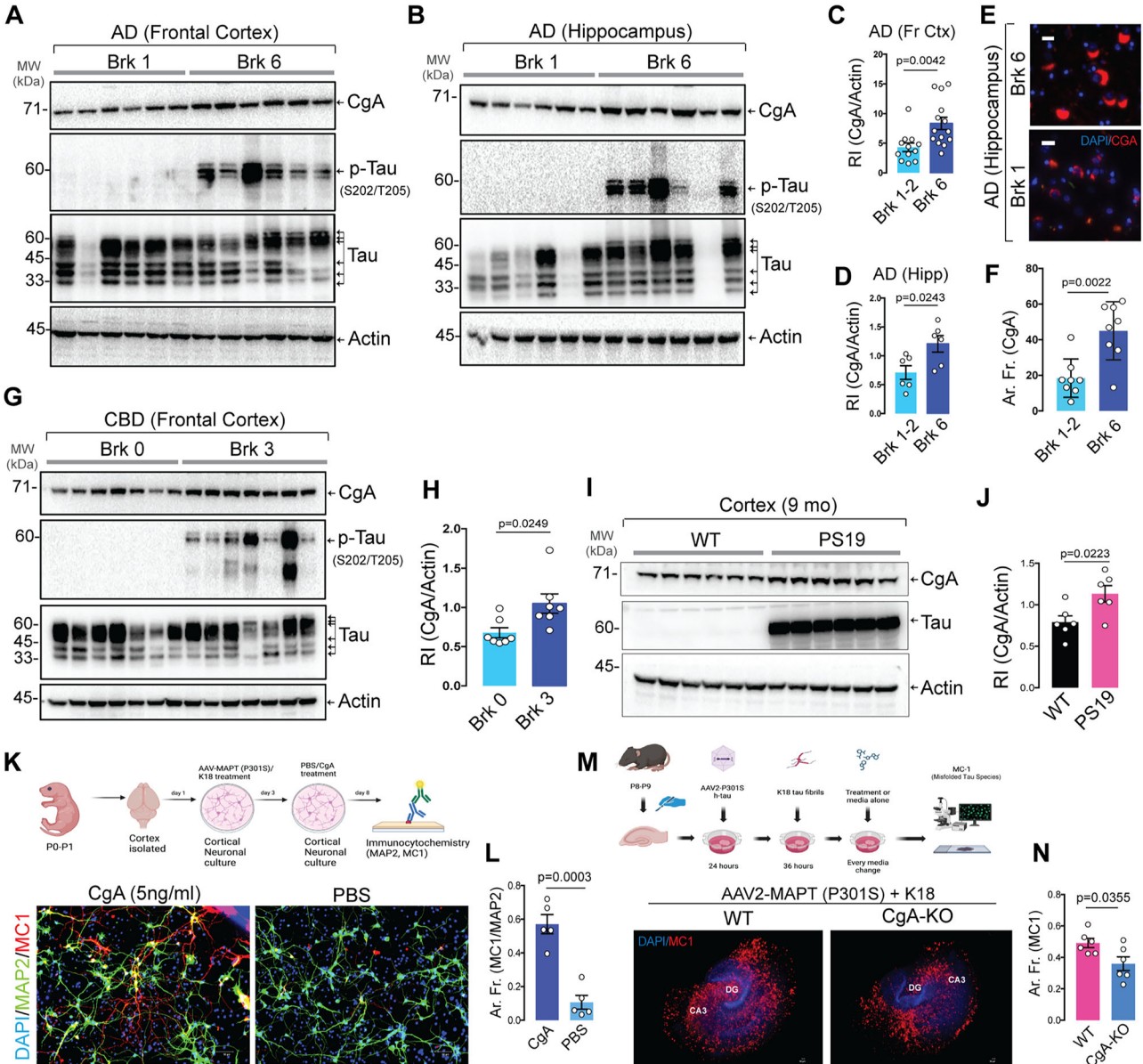

**Fig. 1 | Augmented CgA protein levels in AD and CBD patient samples and PS19 transgenic mice. A** Representative Western blot (WB) of postmortem frontal cortex extracts showing levels of CgA, p-Tau, total Tau and Actin in six Braak stage 6 (Brk 6) and Braak stage 0-2 (Brk 0-2) postmortem patient frontal cortex lysates. **B** Representative WB of hippocampus extracts showing levels of CgA, p-Tau, total Tau and Actin in Brk6 and Brk (0-1) hippocampus lysates. **C** Quantification of WB in (**A**) (Brk 0-2, *n* = 12; Brk6, *n* = 14) (t = 3.19, df = 22.32). **D** Quantification of WB in (**B**) (Brk 0-2, *n* = 6; Brk6, *n* = 6) (t = 2.665, df = 9.49). **E** Representative immuno-fluorescence (IF) images showing CgA signal in the hippocampus of Brk 6 and Brk 1-2 samples. Scale bar = 20 μm. **F** ImageJ quantification of the immunoreactive CgA in the hippocampus shown as area fraction (%) of total hippocampus (*n* = 8 per group) (t = 3.853, df = 12.15). **G** WB of frontal cortex lysates of CBD and normal control patients showing levels of CgA, pTau, Tau and actin. **H** Quantification of CgA levels in WB shown in (**G**), normalized to actin levels (*n* = 7 per group)

(t = 2.687, df = 9.028). **I** WB of frontal cortex lysates of WT (*n* = 6) and PS19 (*n* = 6) mice for CgA, Tau and actin. **J** Quantification of WB shown in **I** for CgA levels, normalized to actin levels (*n* = 6 per group) (t = 2.728, df = 9.455). **K** Schematic (Created in BioRender[81]) showing the experimental plan of CgA treatment in neuron and representative image of MAP2 and MC1 staining after CgA treatment. Scale bar = 200 μm. **L** Quantification of MAP2 and MC1 staining after CgA treatment. (t = 6.623, df = 7.284). **M** Schematic of organotypic slice culture (OTSC) generation, treatment and imaging (Created in BioRender[82]). Representative IF images using MC1 antibody showing misfolded Tau species in organotypic hippocampal slices from WT (top) and CgA-KO (bottom) mice transduced with AAV tau P301S and treated with K18 fibrils. Scale bar = 200 μm. **N** ImageJ quantification of MC1 area fraction from images as represented in (**M**) (*n* = 6), t = 2.49, df = 8.635. P-values were calculated using unpaired two-tailed T-test with Welch's correction. Data are presented as mean values +/− SEM. Source data are provided as a Source Data file.

To quantify the difference in the level of seeding-competent Tau seeds between PS19 and CgA-KO/PS19 cortex, we employed the well-established Förster resonance energy transfer (FRET) assay in HEK 293 T cells expressing Tau-RD fused to CFP/YFP[50]. Lower FRET signal was found in crude brain lysates of CgA-KO/PS19 mice compared to PS19 mice, indicating lower Tau seeding capacity (Fig. 2G, H). We further investigated CgA's impact on Tau seeding and spreading by

stereotaxically injecting synthetic K18 Tau fibrils into the dentate gyrus (DG) of three-month-old PS19 and CgA-KO/PS19 mice[47] (Fig. 2I). Fibril-induced Tau pathology spread to the CA3 region of the contralateral hippocampus was evident in PS19 mice 6 weeks post-injection, but minimal in CgA-KO/PS19 mice, suggesting that CgA promotes Tau spread (Fig. 2J, K, S-Fig. 3M, N). Our control experiment with PBS injection showed no MC1+ aggregates at this age (S-Fig. 3O−P).

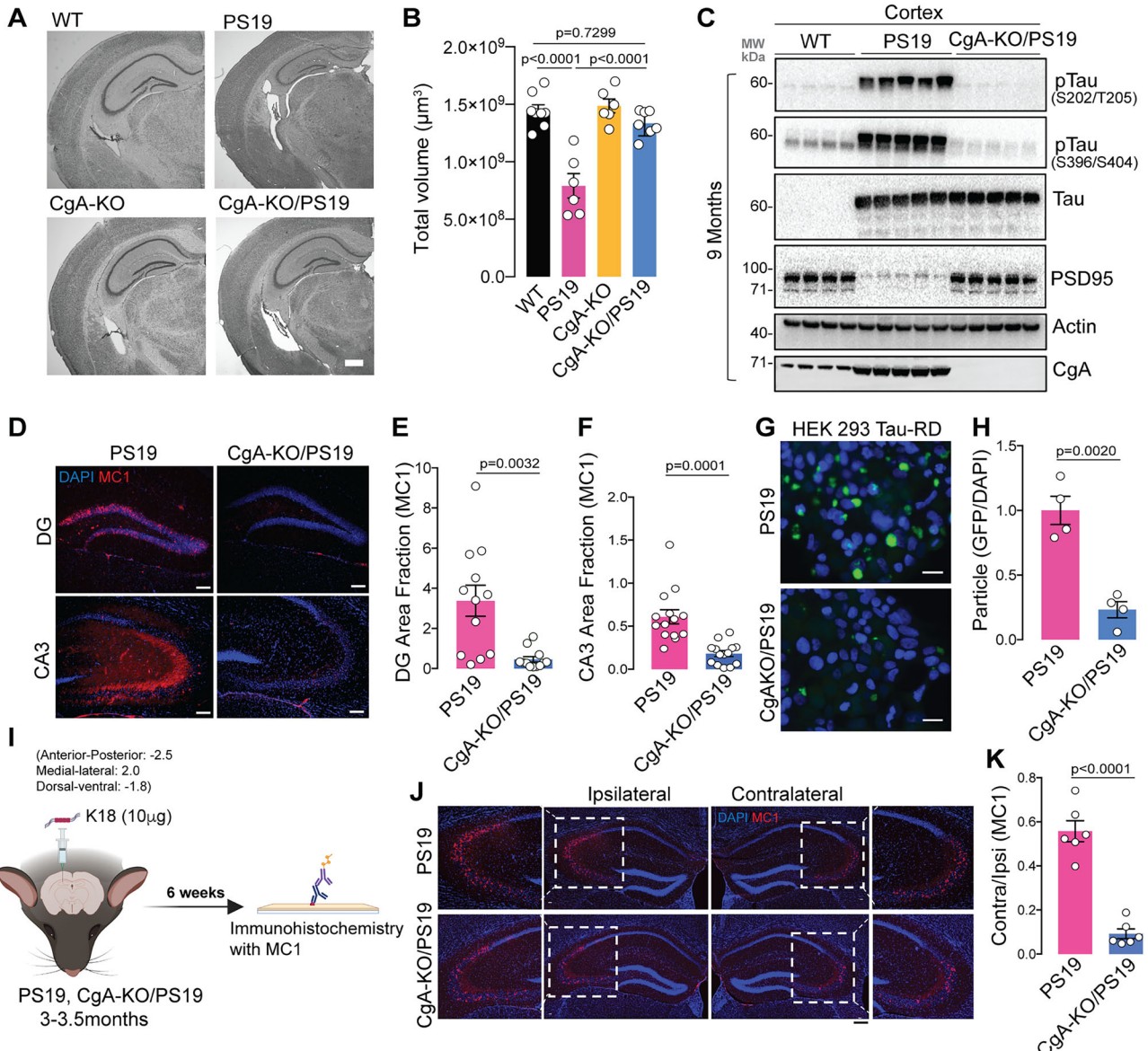

**Fig. 2 | CgA deficiency reduces tauopathy. A** Representative Nissl staining of mouse brain section of WT, PS19, CgA-KO and CgA-KO/PS19 littermate mice of 9-10 months of age. Scale bar = 200 μm. **B** Estimated volume of the hippocampus of 9-10-months-old WT ($n = 8$), PS19 ($n = 6$), CgA-KO ($n = 6$) and CgA-KO/PS19 mice ($n = 7$). (One-way Anova, Sidak's Multiple comparison test, $F_{3, 23} = 1.969$, $P < 0.0001$). **C** WB of cortex RIPA lysates of WT ($n = 4$), PS19 ($n = 5$) and CgA-KO/PS19 ($n = 5$) mice showing levels of pathogenic phosphorylated Tau (pTau), total Tau, PSD95, actin and CgA. **D** Representative IF images showing the levels of MC1 labeled misfolded Tau species in the Dentate Gyrate (DG) and CA3 region of hippocampus. Scale bar = 100 μm. **E** ImageJ quantification of MC1+ misfolded Tau species revealed by IF in DG ($n = 12$) (Unpaired two-tailed t-test with Welch's correction, $t = 3.696$, df = 11.71). **F** ImageJ quantification of MC1+ misfolded Tau species revealed by IF in CA3 ($n = 14$) (Unpaired two-tailed T-test with Welch's correction, $t = 4.836$, df = 17.31).

**G** Fluorescence images of HEK293 cells expressing TauRD-GFP transfected with the brain extracts of PS19 (top) or CgA-KO/PS19 mice. Scale bar = 50 μm. **H** ImageJ quantification of Misfolded Tau species (FRET+) as shown in **G**, normalized to DAPI counts ($n = 4$ per group) (Unpaired two-tailed T-test with Welch's correction, $t = 6.112$, df = 4.753). **I** Schematics describing the Tau fibrils (K18/PL)-induced spreading assay (Created in BioRender[83]). **J** Representative IF images using MC1 antibody showing spreading of misfolded Tau species from the site of injection (ipsilateral) to the opposite site (contralateral) of the hippocampus. A zoomed-in view of the aggregate-rich region is shown in each case. Scale bar = 200 μm. **K** ImageJ quantification of misfolded Tau species as a ratio of MC1+ area in ipsilateral hippocampus relative to the contralateral hippocampus as in (**J**) ($n = 6$ mice per group) (Unpaired two-tailed T-test with Welch's correction, $t = 8.814$, df = 7.042). Data are presented as mean values +/− SEM. Source data are provided as a Source Data file.

Furthermore, IHC staining with Myc and MC1 showed that K18 fibril induced Tau spreading pathology is from endogenous Tau aggregation rather than the injected K18 fibrils themselves (S-Fig. 3Q–R).

We performed ultrastructural analysis via transmission electron microscopy (TEM) and found amplified synapse numbers (marked by arrows) in the hippocampus of CgA-KO/PS19 mice compared to PS19 mice (S-Fig. 4A, B), consistent with increased PSD95 levels in the cortical and hippocampal lysates (Fig. 2C; S-Fig. 3F, G, J). Notably, PS19 mice had fewer synaptic vesicles (SVs) in their presynaptic neurons

compared to WT and CgA-KO/PS19 mice (S-Fig. 4C, D). Collectively, these findings suggest that CgA promotes neurodegeneration in Tau-induced pathogenesis.

## CgA depletion improves cognitive function and longevity of PS19 mice

We conducted behavioral assessments on four genotypes of mouse littermates (WT, CgA-KO, PS19, and CgA-KO/PS19). In the Morris water maze (MWM) test, during the training phase, PS19 mice traveled

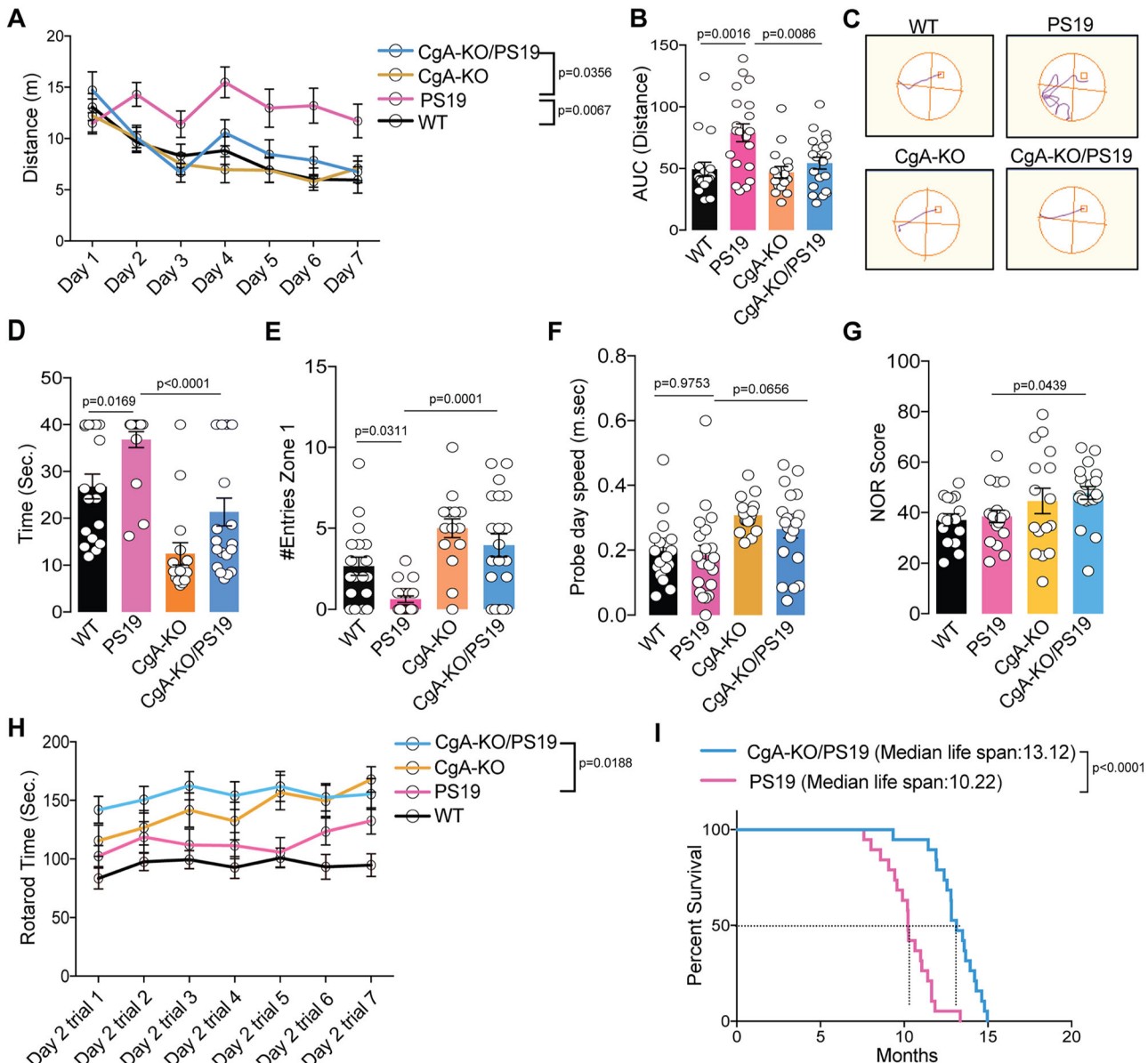

**Fig. 3 | Ablation of CgA improved spatial learning and memory, and extended the life span of PS19 mice. A** Distance traveled each day to reach the platform during seven days training trials of the Morris Water-Maze Test. CgA-KO/PS19 (*n* = 20), CgA-KO (*n* = 16), PS19 (*n* = 19) and WT (*n* = 18). p-value was calculated using Two-way ANOVA ($F_{18, 420}$ = 3.013, P < 0.0001) Tukey's Multiple comparison test. **B** Area Under the Curve (AUC) for the distance traveled during the training period. P-value was calculated using One-way ANOVA ($F_{3, 70}$ = 43.76, P < 0.0001) Dunnett's Multiple comparison test. **C** Probe trial performed 24 hr after the 7th day of training for all the mice. The swimming paths in the probe trial of representative mice of all four groups are shown (the square marks the location of the platform during the goal acquisition trials). **D** Time taken (latency) to reach the target platform. CgA-KO/PS19 (*n* = 20), CgA-KO (*n* = 16), PS19 (*n* = 19) and WT (*n* = 18). P-value was calculated using One-way ANOVA ($F_{3, 69}$ = 16.14, P < 0.0001) Sidak's Multiple comparison test. **E** Number of entries in the zone where the platform was present (correct zone). CgA-

KO/PS19 (*n* = 20), CgA-KO (*n* = 16), PS19 (*n* = 19) and WT (*n* = 18). P-value was calculated using One-way ANOVA ($F_{3, 69}$ = 11.5, P < 0.0001) Sidak's Multiple comparison test. **F** Swimming speed of all the mice during the probe trial day. CgA-KO/PS19 (*n* = 20), CgA-KO (*n* = 16), PS19 (*n* = 20) and WT (*n* = 18). P-value was calculated using One-way ANOVA ($F_{3, 70}$ = 4.99, P = 0.0034) Sidak's Multiple comparison test. **G** Novel Object Recognition scores of CgA-KO/PS19 and PS19 mice. [WT (*n* = 15), CgA-KO (*n* = 16), PS19 (*n* = 19) and CgA-KO/PS19 (*n* = 20)]. P-value was calculated using Kruskal-Wallis test for NOR. **H** Time before falling in the Rotarod test for all four mice groups [WT (*n* = 18), CgA-KO (*n* = 16), PS19 (*n* = 20) and CgA-KO/PS19 (*n* = 20)]. P-value was calculated using Two-way ANOVA ($F_{18, 420}$ = 1.328, P = 0.1656) Turkey's Multiple comparison test. **I** Longevity of PS19 and CgA-KO/PS19 mice shown as a Kaplan-Meier plots (*n* = 20 per group). P-value was calculated using Gehan-Breslow-Wilcoxon test (Chi square = 23.86, df = 1). Data are presented as mean values +/− SEM. Source data are provided as a Source Data file.

prolonged distances and durations before reaching the platform and exhibited lack of improvement over the seven-day training period, compared to WT and CgA-KO mice (Fig. 3A–C; S-Fig. 3A), indicating impaired spatial learning in PS19 mice. In contrast, CgA-KO/PS19 mice exhibited a complete rescue of the learning deficit (Fig. 3A–C; S-Fig. 5A). In the probe trial, PS19 mice took longer time and fewer entries to enter the target quadrant than WT mice, indicative of

memory impairment, whereas CgA-KO/PS19 mice performed indistinguishably from the WT (Fig. 3D, E). Notably, both body weight and swimming speed remained comparable between PS19 and CgA-KO/PS19 mice (Fig. 3F; S-Fig. 5B), indicating that the reduced latency in CgA-KO/PS19 mice was not due to increased speed. In the novel object recognition (NOR) test, CgA-KO/PS19 mice showed increased preference towards the new object compared to PS19 mice, indicative of

improved short-term memory (Fig. 3G). Additionally, the rotarod test revealed significantly improved motor function in CgA-KO/PS19 mice across multiple trials (Fig. 3H). Collectively, these behavioral tests showcased the restoration of learning and memory-related activities upon CgA depletion in PS19 mice. Notably, the CgA-KO group outperformed the WT cohort in all behavioral tests, aligning with the hypothesis that depletion of CgA rejuvenates mice beyond their biological age.

We next determined the lifespan of PS19 and CgA-KO/PS19 mice. The median lifespan of CgA-KO/PS19 mice extended to 13.22 months, compared to 10.12 months in PS19 mice (Fig. 3I), indicating a remarkable ~30% increase in lifespan upon CgA depletion. Notably, 57% ($n = 8$) of CgA-KO/PS19 mice exhibited relatively healthy lives until 12 months of age without discernible disease symptoms. Moreover, 23.6% ($n = 5$) of CgA-KO/PS19 mice survived until 14.5 months, in contrast with the PS19 mice, of which none had survived by this age. Collectively, these findings underscore the benefit of CgA depletion in the PS19 mouse model in improving their cognitive function and mitigating premature mortality.

## CgA depletion partially protects glial activity in PS19 mice

Next, we determined the potential role of CgA depletion in attenuating neuroinflammation, a key contributor to neurodegeneration and subsequent brain atrophy in PS19 mice[51]. We measured the area fraction of glial cell markers, specifically CD68 for microglia and GFAP for astrocytes, using IHC. Strikingly, the area fraction of CD68 in CA1 and CA3 region of the hippocampi of CgA-KO/PS19 mice was significantly lower compared to their PS19 littermates (Fig. 4A–D). Similar reduction in Iba-1 immunofluorescence was observed (Fig. 4E, F). Simultaneously, CgA-KO/PS19 mice exhibited reduced levels of proinflammatory cytokines including Interleukin-1 beta (IL-1β) (Fig. 4G), Interleukin-6 (IL-6) (Fig. 4H), Tumor Necrosis Factor-alpha (TNFα) (Fig. 4I), C-X-C Motif Chemokine Ligand 2 (CXCL2) (Fig. 4J), C-X-C Motif Chemokine Ligand 10 (CXCL10) (S-Fig. 6A), C-X-C Motif Chemokine Ligand 1 (CXCL1) (S-Fig. 6B), C-C Motif Chemokine Ligand 2 (CCL2) (S-Fig. 6C), and C-C Motif Chemokine Ligand 3 (CCL3) (S-Fig. 6D) in cortical extracts compared to PS19 mice. We further found that the plasma proinflammatory cytokine/chemokine levels such as TNFα (S-Fig. 6E), IL-6 (S-Fig. 6F), CXCL10 (S-Fig. 6G), CXCL1 (S-Fig. 6H), CXCL2 (S-Fig. 6I), C-C Motif Chemokine Ligand 2 (CCL2) (S-Fig. 6J), and CCL3 (S-Fig. 6K) were higher in PS19 mice at both three and nine months of age compared to CgA-KO/PS19 mice. Therefore, CgA depletion attenuates the systemic inflammatory response in PS19 mice. On the other hand, the anti-inflammatory cytokine IL-10 level was higher in CgA-KO/PS19 than in PS19 mice (S-Fig. 6L). On the other hand, Glial Fibrillary Acidic Protein (GFAP) levels showed a trend of reduction in CgA-KO/PS19 mice compared to PS19 mice (S-Fig. 6M, N).

## CgA depletion alters the adrenergic signaling in PS19 mice

To elucidate the mechanism by which CgA depletion mitigates the pathogenesis in PS19 mice, we conducted bulk RNA sequencing on the cortices of three mouse genotypes (WT, PS19, and CgA-KO/PS19; $n = 5$ or 6 in each group; 3 males and 2-3 females) (Fig. 5A). Depletion of CgA in CgA-KO/PS19 genotype was again confirmed by absence of CgA mRNA (S-Fig. 7A). In comparison of WT and PS19, 869 genes were expressed lower in WT than in PS19 (Fig. 5B), while 1,193 genes showed significantly higher expression (log2(fold change) > 1.5, FDR < 0.01) (Fig. 5C). Importantly, 417 of the upregulated genes (34.9%) in WT mice were similarly upregulated in CgA-KO/PS19 compared to PS19 mice (Fig. 5D). Gene ontology analyses of this cluster revealed significant enrichment in pathways involved in: (i) neuropeptide signaling pathway, (ii) adenylate cyclase-modulating G-protein-coupled receptor (GPCR) signaling pathway, (iii) axoneme assembly, (iv) regulation of membrane potential, (v) regulation of monoatomic ion transmembrane transport, (vi) catecholamine transport, and (vii) adenylate cyclase-inhibiting GPCR signaling pathway (Fig. 5E). Like upregulated genes, 293 of the downregulated genes (33.7%) in WT were similarly downregulated in CgA-KO/PS19 compared to PS19 mice (Fig. 5F). Gene ontology analyses of this cluster revealed significant enrichment in pathways involved in: (i) potassium ion transmembrane transport, (ii) regulation of monoatomic ion transmembrane transport, (iii) exocytosis, (iv) adenylate cyclase-activating GPCR signaling pathway, (v) Ca(2+)/calmodulin-dependent protein kinase (CaMKK)-AMP-activated protein kinase (AMPK) signaling cascade, (vi) adenylate cyclase-modulating GPCR signaling pathway, and (vii) glucocorticoid biosynthetic process (Fig. 5G). Given CgA's role in catecholamine storage regulation[52] and the influence of catecholamines on GPCR signaling[53], we focused on these pathways as they are most likely mediating the effect of CgA in Tauopathy-mediated neurodegeneration. Specifically, the Adra1 family genes ($\alpha_{1A}$, $\alpha_{1B}$, $\alpha_{1D}$) were upregulated, while the Adra2 family genes ($\alpha_{2A}$, $\alpha_{2B}$, $\alpha_{2C}$) were downregulated in PS19 compared to WT and CgA-KO/PS19 (Fig. 5A; S-Fig. 7B), while no significant change was found in β-adrenergic receptors (Adrb) across all the groups. RT-qPCR further confirmed altered gene expression of *Adra1* and *Adra2* in PS19, which were restored upon CgA deletion (S-Fig. 7C–F). Consistently, analysis of publicly available RNAseq data from control and AD patients showed altered expression of *Adra1a* (S-Fig. 7G, H) and *Adra2a* (S-Fig. 7I, J) in Braak stages V–VI compared to Braak stages 0–I in parahippocampal gyri, indicating a correlation with disease progression (RNAseq data obtained from the Mount Sinai Brain Bank [MSBB] study). No significant change in expression of *Adra1b* and *Adra2b* was observed in MSBB data sets, indicating species-specific effects. Functional analysis highlighted the interconnected role of Adra signaling in various cellular functions (catecholamine transport, neuropeptide signaling, second messenger signaling, and potassium-ion transport), underscoring its significance in Tau pathology (S-Fig. 7J). Given previous reports on the Adra1 aggravating the disease pathology in an Aβ mouse model[54], we hypothesized that significant alteration in the expression of Adra1 and Adra2 receptor classes and their downstream signaling cascades mediates the effects of CgA in Tau pathogenesis.

Verification of protein levels for Adra1b and Adra2b in the cortex aligned with transcript levels. WB analysis demonstrated significantly higher Adra1b and lower Adra2b levels in the cortical extracts of PS19 mice compared to WT and CgA-KO/PS19 mice, while Adrb2 levels remained unchanged (Fig. 5H–J). IHC confirmed higher Adra1b protein levels in the DG and CA3 (mossy fibers) of PS19 mice compared to WT and CgA-KO/PS19 (Fig. 5K, L). Notably, the increase in the area fraction of Adra1b is negatively correlated with the hippocampal volume (Fig. 5M). Collectively, these findings suggest that upregulated Adra1 and compromised Adra2 signaling underlies CgA's detrimental effect in Tauopathy brain.

To investigate the spatial relationship between CgA, Adra1b and Tau, we performed IHC with CgA, Adra1b, and MC1 antibodies (S-Fig. 8). Adra1b was predominantly localized in axons, where it was associated MC1+ misfolded Tau species (S-Fig. 8B–D). In contrast, CgA was observed as puncta in the soma of CA2 and CA3 regions of the hippocampus (S-Fig. 8C, D), where it partially colocalized with MC1+ misfolded Tau species (S-Fig. 8C, D, **white arrows**). Notably, there is minimal colocalization of CgA with Adra1b, suggesting that the effect of CgA on Adra1b is likely indirect.

## Aberrant Adra1 signaling primes Tau pathology in PS19 mice

Along with the altered expression of Adra1, we also observed a downregulation of phosphodiesterases (PDEs), including PDE1, PDE3, and PDE9, in PS19 mice compared to WT and CgA-KO/PS19 mice (S-Fig. 9A). Since PDEs degrades cAMP - a key second messenger in the adrenergic system, we measured cAMP levels in WT, PS19 and CgA-KO/PS19 mice. As anticipated, PS19 mice exhibited significantly higher

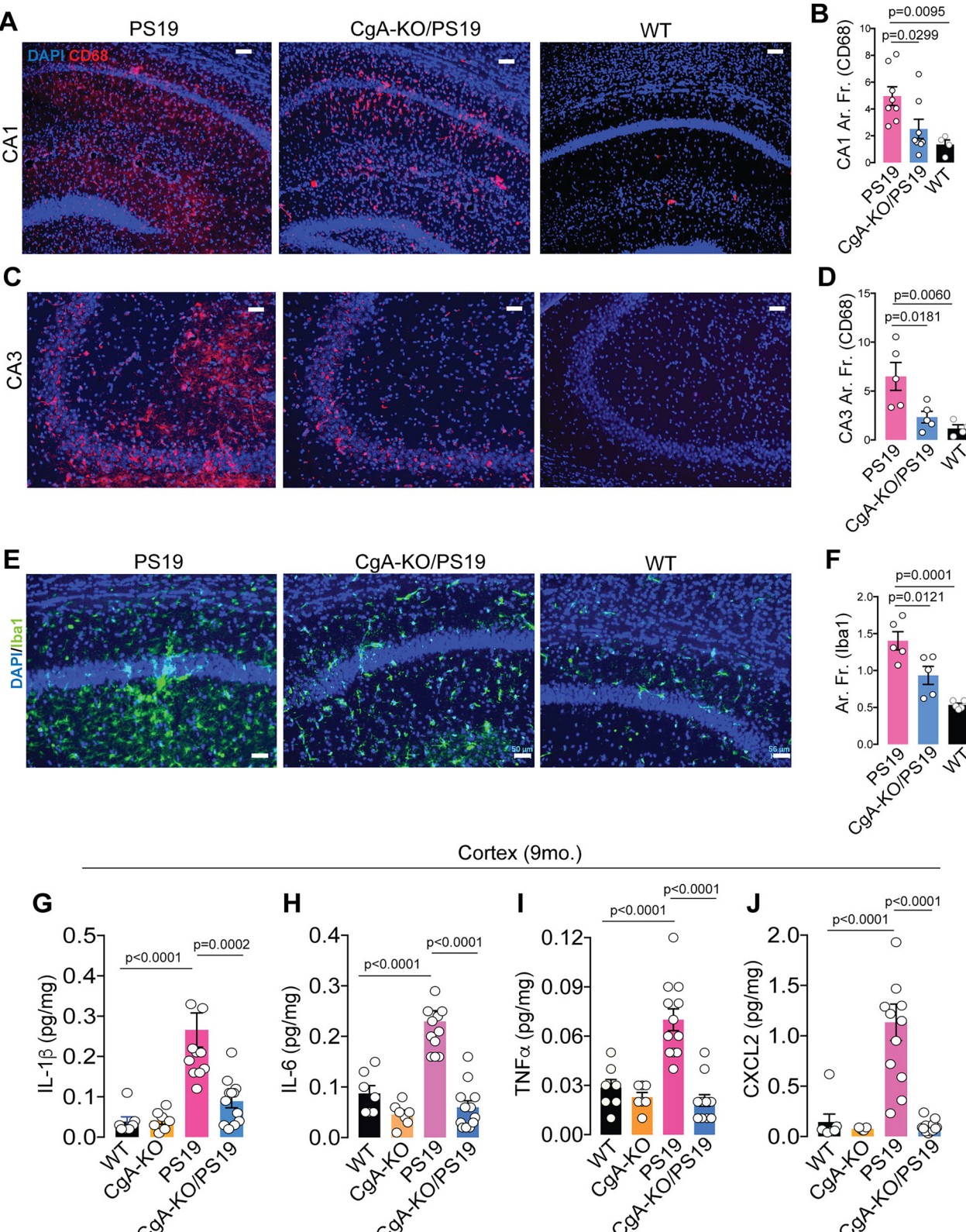

cortical cAMP levels compared to WT and CgA-KO/PS19 mice (S-Fig. 9B). Consistently, we observed elevated cortical cAMP levels in AD Braak stage VI compared to Braak stage 0–II samples (S-Fig. 9C). These findings suggest that the upregulation of Adra1 and concurrent downregulation of Adra2 causes an imbalance signaling cascade with sustain elevated cAMP levels, potentially contributing to exacerbated Tau toxicity.

To directly determine the effect of Adra1 signaling on Tau, we treated OTSC with an Adra1 antagonist (Prazosin, PR) or agonist (Phenylephrine, PE) (Fig. 6A). WT and CgA-KO OTSC transduced to express P301S human Tau (AAV2-P301S-hTau) were seeded with K18 Tau fibrils and treated with PR and PE, respectively (Fig. 6B). Three weeks post-treatment, WB analysis revealed significantly lower levels of pTau (S202, S396, and S404) in PR-treated WT slices compared to

**Fig. 4 | CgA-KO/PS19 mice exhibit decreased microgliosis compared to PS19 mice. A** Representative immunofluorescence (IF) images showing the levels of CD68 in the hippocampi (CA1 region) of PS19, CgA-KO/PS19 and WT mice. Scale bar = 50 μm. **B** Quantification of the immunoreactive CD68 in CA1 as percentage of total CA1 area from IF images as shown in (**A**). PS19 ($n$ = 8), CgA-KO/PS19 ($n$ = 8) and WT ($n$ = 4). P-value was calculated using One-way Anova ($F_{2, 17}$ = 0.6355, P = 0.5418) Dunnett's multiple comparison test. **C** Representative IF images showing the levels of CD68 in the hippocampi (CA3 region) of PS19, CgA-KO/PS19 and WT mice. Scale bar = 50 μm. **D** Quantification of the immunoreactive CD68 in CA1 as percentage of total CA1 area from IF images as shown in (**C**) PS19 ($n$ = 5), CgA-KO/PS19 ($n$ = 5) and WT ($n$ = 4). P-value was calculated using One-way Anova ($F_{2, 11}$ = 4.13, P = 0.0459)

Dunnett's multiple comparison test. **E** Representative IF images showing the levels of Iba1 in the hippocampi of PS19, CgA-KO/PS19 and WT mice. Scale bar = 50 μm. **F** Quantification of the immunoreactive Iba as percentage of total area from IF images as shown in **E**. PS19 ($n$ = 5), CgA-KO/PS19 ($n$ = 5) and WT ($n$ = 5). P-value was calculated using One- way Anova ($F_{2, 12}$ = 2,395, P = 0.1333) Dunnett's multiple comparison test. **G** Levels of cortical IL-1β (**G**), IL-6 (**H**), TNFα (**I**) and CXCL2 (**J**) in PS19 ($n$ = 12), CgA-KO/PS19 ($n$ = 12), WT ($n$ = 7) and CgA-KO ($n$ = 7). P-values were calculated using One-way ANOVA (**G** $F_{3, 34}$ = 14.35, P < 0.0001, **H** $F_{3, 34}$ = 28.6, P < 0.0001, **I** $F_{3, 34}$ = 22.93, P < 0.0001, **J** $F_{3, 34}$ = 21.34, P < 0.0001) Tukey's Multiple Comparison test. Data are presented as mean values +/− SEM. Source data are provided as a Source Data file.

DMSO-treated controls (Fig. 6B, S-Fig. 10A, B) and higher levels of pTau (S202, S396, and S404) in PE-treated slices compared to PBS-treated controls (Fig. 6B). At baseline, IHC analysis using the MC1 antibody also showed abundant misfolded Tau species in DMSO-treated WT slices (Fig. 6C, S-Fig. 10C) and fewer misfolded Tau species in PBS-treated CgA-KO slices (Fig. 6D). While PR reversed (decreases) MC1 aggregation in WT slices (Fig. 6C), Tau aggregation was reversed (increased) in PE-treated CgA-KO slices (Fig. 6D). These data suggest that Adra1-signaling promotes the development of Tau pathology. Dose-dependent decrease in Tau phosphorylation (S-Fig. 10D) and Tau aggregation (S-Fig. 10E, F) by PR in CgA-KO slices implicate that the inhibition of Tau pathology by CgA-KO is, at least partially, mediated by decreased Adra1 signaling. Likewise, PE caused dose-dependent increase in Tau phosphorylation (S-Fig. 10G) and Tau aggregation (S-Fig. 10H, I) in WT slices, reinforcing adrenergic signaling in Tau pathology.

Since PR can cross the blood brain barrier (BBB)[55], we assessed its potential to reduce Tau pathology in PS19 mice. Six-month-old PS19 mice were treated with PR via weekly (two times a week) intraperitoneal injection for 12 weeks, followed by WB (Fig. 6E–G) and IHC analysis (Fig. 6H). Remarkably, PR treatment significantly reduced levels of pTau (Fig. 6E–G) and misfolded Tau species (Fig. 6H, S-Fig. 10J) in the hippocampus. Collectively, these results strongly suggest that Adra1 activation drives Tau pathology.

### Elevated EPI-Adra1 signaling drives Tau phosphorylation and aggregation

Since EPI and NE both are endogenous agonist of adrenergic receptors, the above results suggest that they both might be involved in Adra1-induced disease progression. Therefore, we assessed EPI and NE levels in the prefrontal cortex (Fig. 7A; S-Fig. 11A), hippocampus (Fig. 7B; S-Fig. 11B) and cerebrospinal fluid (CSF) (Fig. 7C; S-Fig. 11C) from AD patients. Elevated EPI levels were observed in Braak VI AD pre-frontal cortex (Fig. 7A), hippocampus (Fig. 7B), and CSF (Fig. 7C) samples compared to Braak 0-2 samples. No significant change in NE level was detected in prefrontal cortex (S-Fig. 11A) and hippocampus (S-Fig. 11B), whereas CSF levels of NE was decreased (S-Fig. 11C). Like AD samples, CBD prefrontal cortical samples also showed higher EPI levels than in normal controls (Fig. 7D). NE levels did not differ between CBD patients and controls samples (S-Fig. 11D). These results are consistent with prior reports showing elevated EPI concentrations in the CSF of AD and dementia patients, correlating with disease progression, while the difference in NE levels between patients and nonpatients was not statistically significant[56]. Strikingly, both EPI and NE levels were elevated in the cortices of PS19 mice compared to WT, CgA-KO, and CgA-KO/PS19 mice (Fig. 7E; S-Fig. 11E). Plasma EPI levels were similarly changed in PS19 mice (S-Fig. 11F), whereas NE levels were elevated in PS19 (S-Fig. 11G) but unchanged in CgA-KO and CgA-KO/PS19 mice (S-Fig. 9F, G). These findings are in line with the known involvement of CgA in the secretion and/or metabolism of catecholamine[52]. Given that neither EPI nor NE can cross the BBB, these results suggest that CgA might be critical in regulating brain-residential production of

catecholamine. Since EPI levels exhibited consistent increase in PS19 mice and in CBD and AD samples, we focused specifically on EPI.

To test the effect of EPI on Tau pathology, we performed EPI treatment on WT OTSC expressing P301S human Tau and seeded with K18 fibrils. WB (Fig. 7F–H) and IHC (Fig. 7I, J) analysis showed that EPI significantly induced Tau phosphorylation (Fig. 7F–H) and aggregation (Fig. 7I, J). To further determine if EPI-induced Tau pathology is dependent on CgA, we treated CgA-KO OTSC with EPI and found that EPI failed to induce Tau phosphorylation (S-Fig. 11H–J) or Tau aggregate formation (S-Fig. 11K, L). These observations underscore the potential contribution of elevated EPI levels to the development of Tau pathology, which is regulated by CgA.

To establish the CgA-EPI-Adra1 signaling axis in regulation of Tau pathogenesis, we previously treated CgA-KO OTSC with EPI (S-Fig. 11H–L) and found no significant increase in Tau pathogenic burden. To further confirm CgA-EPI-Adra1 signaling axis in regulation of Tau pathogenesis, we performed co-treatment of EPI and PR in OTSC (Fig. 7K, L). Remarkably, EPI + PR co-treatment significantly reduced EPI-induced increase in Tau aggregation (Fig. 7L, M) and Tau phosphorylation (S-Fig. 11M–O), indicating that EPI increases Tau pathology via activation of Adra1 signaling.

### Discussion

This study uncovers previously unexplored roles of CgA and EPI in the pathogenesis of Tau in the brains of Tauopathies, including AD, CBD and mouse models (Fig. 8). We discovered that in Tauopathy brains, there are elevated levels of CgA, EPI, and aberrant Adra signaling. Depletion of CgA in P301STau transgenic mice normalized Adra signaling, reduced Tau pathology and ameliorated Tau-mediated neurodegeneration. This highlights the significance of aberrantly elevated EPI signaling as a key contributor to Tau pathogenesis in vivo. These findings align with the clinical observations in CBD and AD patients, where elevated EPI levels have been documented in the CSF[56,57]. Consistently, we observed increased EPI levels in the cortex and hippocampus, along with an increase in Adra1 and a decrease in Adra2 transcripts in AD brains.

Adra1 receptors are intricately linked to G-protein signaling pathways, particularly through Gq, which triggers calcium flux. This calcium influx activates several kinases, including protein kinase C (PKC). Additionally, calcium can activate specific members of the adenylate cyclase family, leading to increased cAMP levels[58]. In contrast, Adra2 receptors, serving as negative regulators in the adrenergic system, induce the inhibitory G protein Gi, blocking adenylyl cyclase and suppressing cAMP production[59]. Activation of Adra2 by EPI in presynaptic neurons blocks their secretion through inhibition of calcium-mediated exocytosis of catecholamine storage vesicles[60]. CST has been reported to inhibit catecholamine secretion from chromaffin cells and hippocampal neurons[15,61]. Therefore, it is likely that the lack of CgA processing into CST and Adra2 downregulation are related and are responsible for enhanced EPI secretion in the brain. This notion is supported by reports that EPI lowers the transcription of the *Adra2a* gene in rat astroglia by acting on Adra1 and Adrb and activating PKC

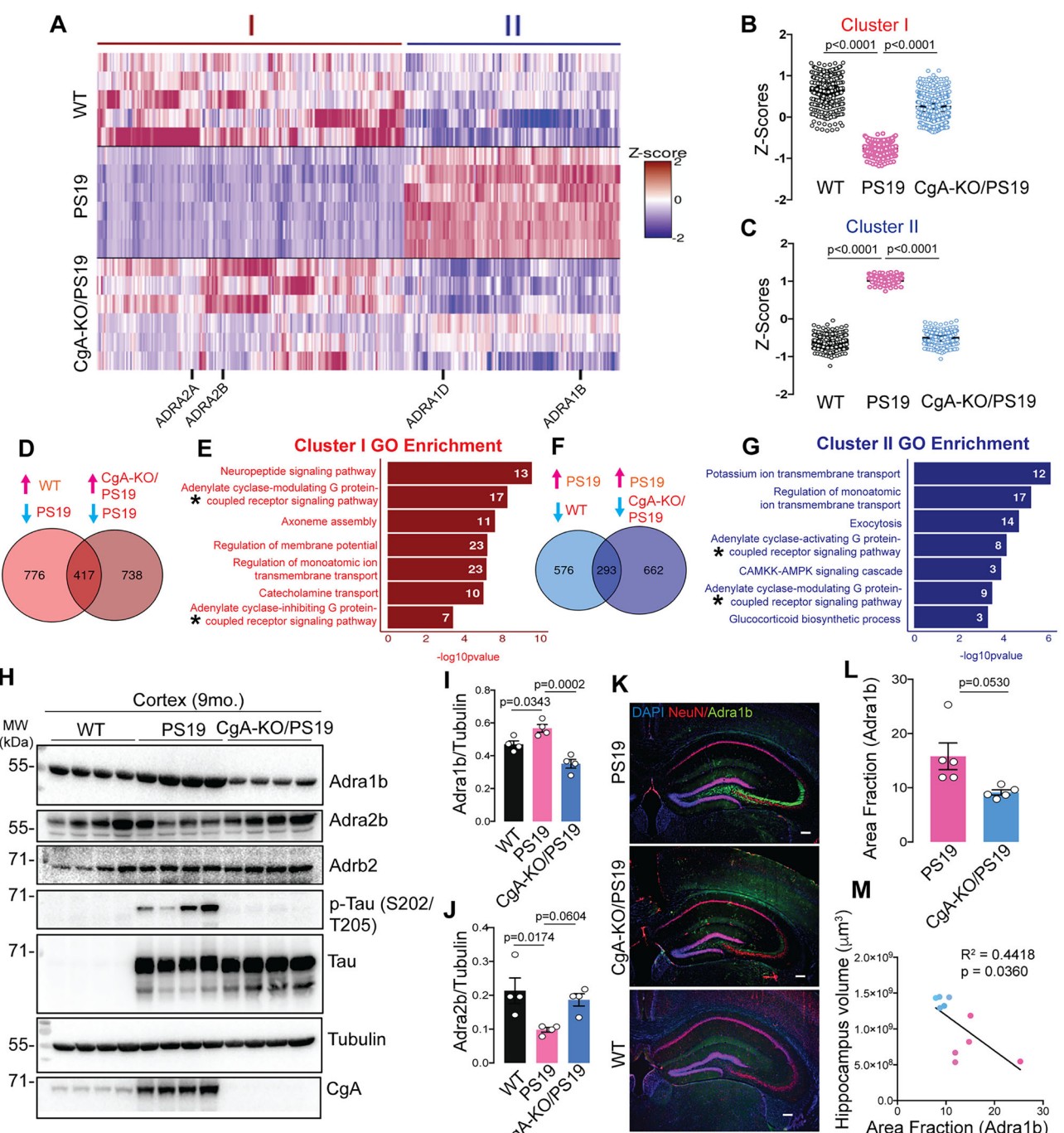

**Fig. 5 | CgA deletion induces reprogramming of adrenergic receptor expression in Tauopathy mice brain. A** Heatmap representation of normalized read Z-scores of shared upregulated or downregulated genes in WT ($n = 5$) and CgA-KO/PS19 ($n = 6$) relative to PS19 ($n = 6$) (log2 fold-change > 0.5, p value < 0.01) from 9-month-old mice cortices. *Adra* gene family showing significant expression changes are highlighted. **B** Z-score of hierarchical clusters I of WT, PS19 and CgA-KO/PS19. P-value was calculated using One-way Anova ($F_{2, 1248} = 163.9$, $P < 0.0001$) Sidak's multiple comparison test. **C** Z-score of hierarchical clusters II of WT, PS19 and CgA-KO/PS19. P-value was calculated using One-way Anova ($F_{2, 876} = 45.91$, $P < 0.0001$) Sidak's multiple comparison test. **D** Venn diagram showing number of upregulated genes as compared in two groups (WT/PS19 and CgA-KO/PS19). Expression of 417 genes overlaps in PS19 vs. WT and PS19 vs. CgA-KO/PS19. **E** Gene ontology analysis for pathways enriched in shared genes in (**D**). Number of genes involved in each pathway are shown in the bar. **F** Venn diagram showing number of downregulated genes as compared in two groups (WT/PS19 and CgA-KO/PS19). Expression of 293 genes overlaps in PS19 vs. WT and PS19 vs. CgA-KO/PS19. **G** Gene ontology analysis for pathways enriched in shared genes in (**E**) Number of genes involved in each

pathway are shown in the bar. **H** Representative WB showing Adra1b, Adra2b and Adrb2 protein levels in three mouse groups, WT ($n = 4$), PS19 ($n = 4$) and CgA-KO/PS19 ($n = 4$). Also shown are the levels of p-Tau (S202/T205), total PS19, CgA and tubulin (loading control). **I** Quantitation showing relative levels of Adra1b in (**H**). WT ($n = 4$), PS19 ($n = 4$) and CgA-KO/PS19 ($n = 4$). P-value was calculated using One-way Anova ($F_{2, 9} = 0.07055$, $P = 0.0004$) Sidak's multiple comparison test. **J** Quantitation showing relative levels of Adra2b in (**H**). WT ($n = 4$), PS19 ($n = 4$) and CgA-KO/PS19 ($n = 4$). P-value was calculated using One-way Anova ($F_{2, 9} = 0.8729$, $P = 0.0004$) Sidak's multiple comparison test. **K** Representative IF images of Adra1b, NeuN and DAPI in the hippocampus slices of PS19, CgA-KO/PS19 and WT mice. Scale bar = 200 μm. **L** Image (**J**) Quantitation of IF images as in (**H**) showing relative levels of Adra1b in PS19 ($n = 5$) and CgA-KO/PS19 ($n = 5$). P-value was calculated using (Unpaired two-tailed T-test with Welch's correction, t = 2.649, df = 4.293).
**M** Pearson correlation between Adra1b area fraction and the hippocampus volumes of PS19 (red dots) and CgA-KO/PS19 (blue dots) mice. Data are presented as mean values +/− SEM. Source data are provided as a Source Data file.

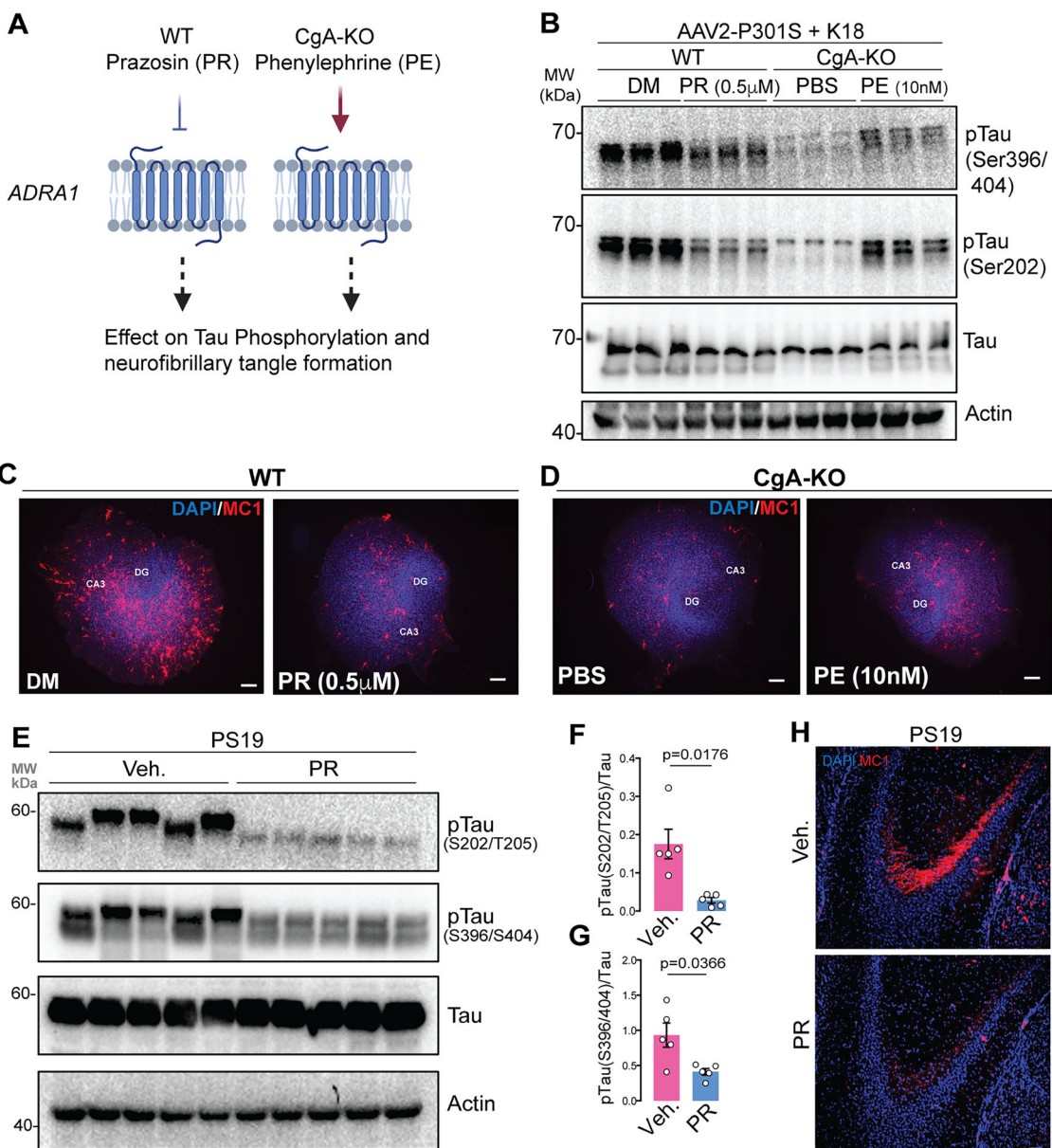

**Fig. 6 | Alpha-1 adrenergic signaling regulates Tauopathy. A** Schematics showing the rationale of Phenylephrine (Adra1 agonist) and Prazosin (Adra1 antagonist) treatment in WT and CgA-KO hippocampal OTSC to access the Tau aggregation ex-vivo (Created in BioRender[84]). **B** Representative WB showing p-Tau, total tau and actin (loading control) from OTSC lysates upon different treatment. WT hippocampus OTSC treated with DMSO ($n=3$) and PR (Prazosin) ($n=3$). CgA-KO hippocampus OTSC treated with PBS ($n=3$) and PE (Phenylephrine) ($n=3$). **C** Representative IF images showing MC1+ misfolded Tau species in WT hippocampus OTSC treated with PR ($n=6$) and DMSO ($n=6$) control. Scale bar = 200 µm. **D** Representative IF images showing MC1+ misfolded Tau species in CgA-KO hippocampus OTSC treated with PBS ($n=6$) and PE ($n=6$). Scale bar = 200 µm. **E** Western Blot of cortex RIPA lysate of Prazosin (PR) treated ($n=5$) and Vehicle (Veh.) treated ($n=5$) PS19 mice showing levels of pTau, Tau and actin. **F, G** Densitometric quantification of p-Tau in PR ($n=5$) and veh. ($n=5$) treated mice. P-value in F and G was calculated using Unpaired two-tailed T-test with Welch's correction ((**F**) t = 3.757, df = 4.267, (**G**) t = 2.924, df = 4.577). **H** Representative IHC image of MC1 staining in PR and veh. treated mice. Scale bar = 50 µm. Data are presented as mean values +/− SEM. Source data are provided as a Source Data file.

and cAMP[62]. Although the reciprocal relationship of expression levels between Adra1 and Adra2 has not been shown in neurons, their presence in all brain cell types suggests EPI might exert its effect in neurons as well. Our observation that heightened Adra signaling is at least partly responsible for Tau pathogenesis aligns with the previous finding that Adra1 antagonist alleviates memory loss in a mouse model (APP23) of AD[63]. Future research will address the precise mechanism of EPI secretion in the brain. Our observations parallel reports in the rhesus monkey model of AD linking increased Ca$^{2+}$, cAMP, p-Tau, and tangles[64]. However, excess NE, not EPI, was attributed to elevated Adra signaling[59]. Although excess NE could be pathogenic, our observations

strongly suggest that the overactive EPI-Adra1 signaling pathway is key to Tau phosphorylation and aggregation in brain regions like the hippocampus and cortex. Collectively, these observations implicate that dysregulation of Adra signaling, including upregulation of Adra1 and down regulation of Adra2, exacerbates Tau pathogenesis.

The concentration of EPI in the brain is ~1000-fold less than in the adrenal medulla. The intriguing question of how brain EPI functions as a neurotransmitter remains unanswered. The exclusive producers of EPI in the brain are the C1 neurons located in the rostral and intermediate portions of the ventrolateral medulla[65]. These neurons express the enzyme phenylethanolamine N-methyltransferase

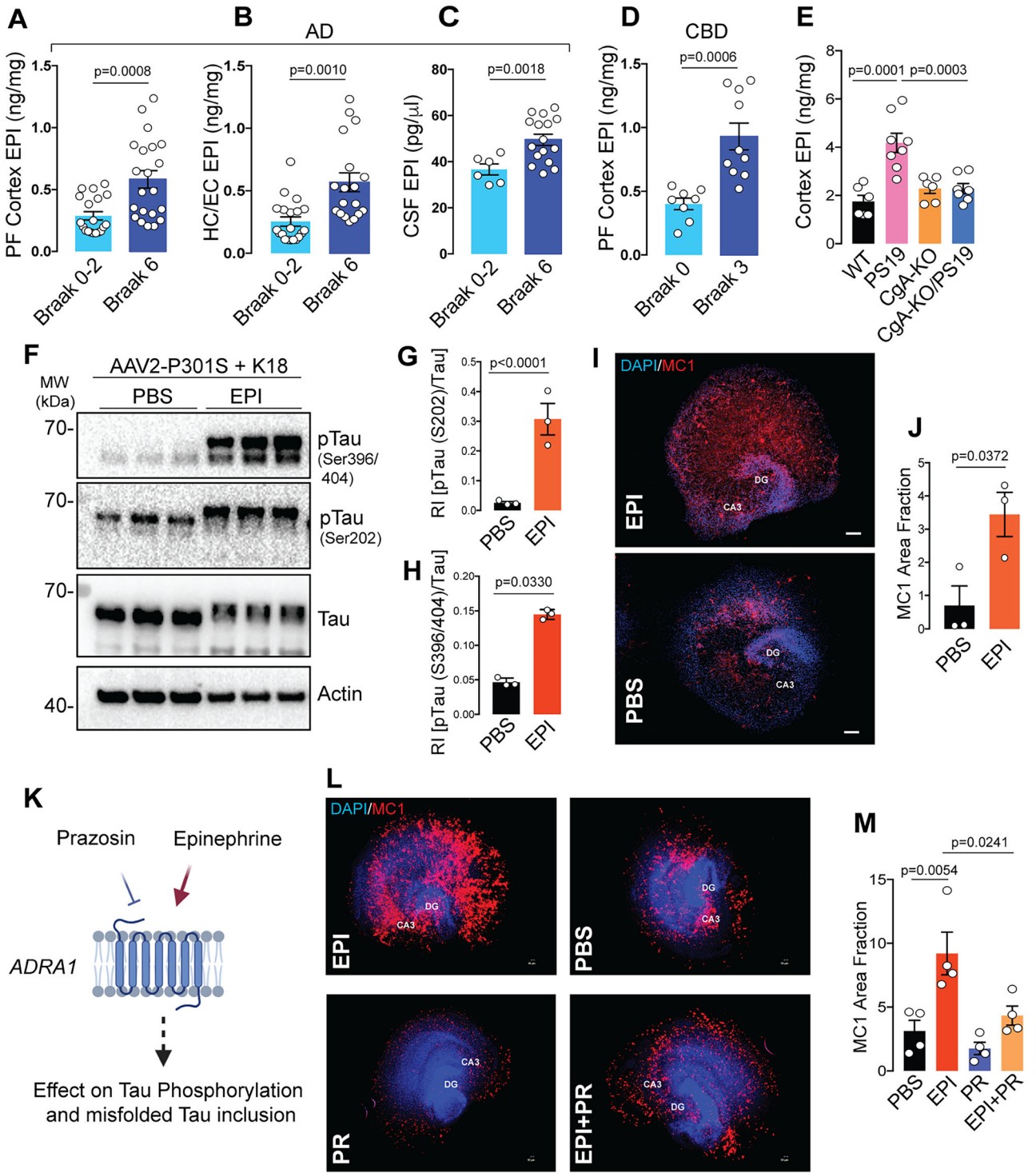

(PNMT)[66], which converts NE to EPI[67]. Increased expression of PNMT and the resultant production of EPI are believed to be responsible for ~28% reduction in C1 neurons in advanced AD[68,69]. EPI containing C1 neurons project to the noradrenergic neurons in the locus coeruleus (LC) - the sole source of NE production in the brain and inhibit the secretion of NE[70] at diverse postsynaptic neurons in the hippocampus and cortex. NE not only contributes to neuroprotection[71] and suppresses neuroinflammation[72,73] but is a major modulator of behavior[74]. These published reports and our observations prompted us to propose that the surplus EPI produced in C1 neurons of Tauopathy brains leads to aberrant NE secretion in postsynaptic noradrenergic neurons,

affecting neuronal health. Thus, C1 EPI might be a pathogenic trigger. Unregulated EPI can also impact glial cells, as they express Adra, potentially triggering inflammation. EPI-induced inflammation has been observed in the lungs, where EPI activates cytokine expression in lung phagocytes[75]. Therefore, unchecked EPI may play a substantial role in microglia activation and neuroinflammation, further contributing to neurodegeneration. Importantly, LC is the brain region affected by Tau pathology at the earliest stage. The C1 nuclei, situated in the medulla oblongata beneath the pons in the brainstem, are responsive to various stressors such as inflammation, hypoglycemia, hypotension, and hypoxia, contributing to the maintenance of

**Fig. 7 | Elevated Epinephrine (EPI) levels in AD, CBD and PS19 transgenic mice samples increases Tauopathy in OTSC through alpha-1 adrenergic signaling.**
**A** Quantification of epinephrine (EPI) levels in the Pre-frontal (PF) cortex of AD patients from Braak stage 0-2 ($n = 19$) and Braak stage 6 ($n = 21$). P-value was calculated using Unpaired two-tailed T-test with Welch's correction, t = 3.748, df = 28.57. **B** Quantification of EPI levels in the hippocampus/Entorhinal (HC/EC) cortex of AD patients from Braak stage 0-2 ($n = 20$) and Braak stage 6 ($n = 18$). P-value was calculated using Unpaired two-tailed t-test with Welch's correction, t = 3.726, df = 24.85. **C** Quantification of EPI levels in the CSF of AD patients from Braak stage 0-2 ($n = 6$) and Braak stage 6 ($n = 16$). P-value was calculated using Unpaired two-tailed t-test with Welch's correction, t = 3.764, df = 15.17. **D** Quantification of EPI levels in the PF cortex of CBD patients from Braak stage 0 ($n = 8$) and Braak stage 3 ($n = 10$) stages. P-value was calculated using Unpaired two-tailed t-test with Welch's correction, t = 4.622, df = 12.17. **E** Quantification of EPI levels in the cortex of 9-months old mice WT ($n = 6$), PS19 ($n = 8$), CgA-KO ($n = 6$), and CgA-KO/PS19 ($n = 7$). P-value was calculated using One-way Anova ($F_{3, 23} = 1.545$, P = 0.2298) Dunnett's multiple comparison test. **F** WB showing p-Tau, total tau and actin (loading control) in hippocampus OTSC lysates upon EPI treatment. **G** Quantification of WB of Tau phosphorylation at S202 relative to total Tau ($n = 3$). P-value was calculated using Unpaired two-tailed t-test with Welch's correction, t = 18.32, df = 3.944. **H** Quantification of WB of Tau phosphorylation at S396/S404 relative to total Tau ($n = 3$). P-value was calculated using Unpaired two-tailed t-test with Welch's correction, t = 5.292, df = 2.025. **I** Representative IF images showing MC1+ misfolded Tau species in hippocampus OTSC treated with EPI (top) or PBS control (bottom). Scale bar = 200 μm. **J** Image J quantification of MC1+ fraction in EPI ($n = 3$) and PBS ($n = 3$) treated hippocampus OTSC. P-value was calculated using Unpaired two-tailed t-test with Welch's correction, t = 3.095, df = 3.938. **K** Schematic of the use of EPI and prazosin (PR) in OTSC to access Tau phosphorylation and neurofibrillary tangle formation (Created in BioRender[84]). **L** Representative images showing MC1 positive pathological Tau conformations in WT hippocampus OTSC treated with EPI, PBS, EPI+PR. Scale bar = 200 μm. **M** ImageJ quantification of MC1+ Tau conformations in OTSC ($n = 4$, per group). P-value was calculated using One-way Anova ($F_{3, 12} = 9.786$, P = 0.0015) Sidak's Multiple Comparisons Test. Data are presented as mean values +/− SEM. Source data are provided as a Source Data file.

physiological homeostasis[65,76]. Given the associations between Tauopathy/AD and inflammation, hyperglycemia, and hypertension and the pivotal role played by C1 neurons in regulating these stresses, it is plausible that EPI from C1 neurons plays a multifaceted role in both health and disease. Future research is necessary to unravel the intricate details of EPI's role in the brain.

We also observed that in Tauopathy brains, CgA and aggregated Tau are partially colocalized, which could initiate non-physiological processes such as diminished axonal transport, EPI storage and release[52,77]. It has been previously shown that the processing of CgA to CST is diminished in hypertensive subjects ($n = 215$) compared to normotensive subjects ($n = 452$)[78]. Therefore, like in hypertension[78], this complex might hinder the processing of CgA into its bioactive peptide CST, which possesses anti-inflammatory properties[16,18], maintains insulin sensitivity[16,17], and inhibits catecholamine release[15,20,21,61], thereby modulates Tau pathogenesis. Further studies are essential to elucidate the intricate and dynamic relationship between CgA, CST and Tau, and how they collectively regulate EPI levels. Subsequent experiments should also clarify the spatiotemporal changes of EPI-Adra axis during disease progression, which will inform future strategies for intervention.

## Methods
### Animals
All studies performed on animals were approved by the UCSD and Veteran Affairs San Diego Institutional Animal Care and Use Committee (IACUC) and conform to relevant National Institutes of Health guidelines. Chromogranin A knockout (CgA-KO) mice on a mixed background (50% 129/SvJ; 50% C57BL/6) were backcrossed to C57BL/6 J mice for 7 generations to get CgA-KO mice in C57BL/6 background[22]. *Cre-loxP* gene-targeting strategy was used to generate congenital and whole body CgA-KO mice. CgA is a unique pro-protein having counter-regulatory domains such as PST and CST. The above CgA-KO mice were backcrossed to B6C3F1/J mice for 4 generations to generate CgA-KO mice in B6C3F1/J background (Mahata Lab). These mice were crossed with PS19 (Jackson Lab) heterozygote mice (in B6C3F1/J background) to generate CgA-KO/PS19 mice. Animals were kept in a 12-hour light/12-hour dark cycle. Mice have access to food and water ad libitum. Mice were fed a regular chow diet (NCD: 14% calorie from fat; LabDiet 5P00).

Mice were euthanized using inhalational isoflurane in compliance with institutional IACUC protocols and the AVMA Guidelines for the Euthanasia of Animals (2020). Animals were placed in an induction chamber pre-filled with 3−5% isoflurane in oxygen (flow rate: 1−2 L/min) and continuously monitored until loss of the righting reflex and absence of response to toe pinch, confirming a surgical plane of anesthesia. Under deep anesthesia, tissues were collected, and euthanasia was completed by exsanguination. All animal procedures were approved by the Institutional Animal Care and Use Committee and conducted in accordance with the ARRIVE guidelines.

### Genotyping
Mice were ear-tagged, and tail-snips were collected. Genomic DNA from tail-snips were isolated using the Accustart Genotyping kit and amplified by PCR with Accustart Geltrack PCR supermix. Primers used for PS19 genotyping are Forward (WT and mutant): 5' TTG AAG TTG GGT TAT CAA TTT GG 3', Reverse (WT): 5' TTC TTG GAA CAC AAA CCA TTT C 3', Reverse (Mutant): 5' AAA TTC CTC AGC AAC TGT GGT 3'. Primers used for *Chga* genotyping are Forward: 5'GTA GCA TGG CCA CTA CCC AG 3' and Reverse: 5' ATC CTT CAG AGC CCC TCC TT 3'.

### Hippocampal slice culture
Organotypic hippocampal slice cultures (OTSC) were performed with hippocampal slices obtained from postnatal day 7-8 pups of WT or CgA-KO mice as described previously (Croft, Noble, 2018). Pups were culled by decapitation, and the hippocampus was quickly dissected in oxygenated artificial cerebrospinal fluid (ACSF). Approximately 18-24 400 μm thick slices were cut using a tissue chopper, and these were cultured in Millicell culture inserts (Millipore) in 6 well plates (6-8 slices per insert). Culture media was changed every 2-3 days, and the slices were harvested after 21 or 28 days. The composition of ACSF was 125 mM NaCl, 2.4 mM KCl, 1.2 mM NaH$_2$PO$_4$, 1 mM CaCl$_2$, 2 mM MgCl$_2$, 25 mM NaHCO$_3$, and 25 mM Glucose. The composition of OTSC culture media was 25 mM HEPES, 133 mM NaCl, 31 mM NaHCO$_3$, 45 mM D-Glucose, 5 mM KCl, 3.3 mM MgSO$_4$, 4.5 mM CaCl$_2$, 1 mM Na$_2$HPO$_4$, 0.5 mM Ascorbic Acid, 1.25 ml Pen-Strep, 2.5 ml Glut-Max, 62.5 ml Heat Inactivated Horse Serum and 82.5μl Insulin. To evaluate tau spreading, organotypic hippocampal slices were fixed in 4% paraformaldehyde (PFA) at room temperature for 4 hours. Following fixation, slices were permeabilized overnight in 1% Triton X-100 in PBS. Antigen retrieval was performed using citrate buffer (pH 6.0) under low pressure for 15 minutes. Slices were then blocked in 20% bovine serum albumin (BSA) with 0.4% Triton X-100 for 3 hours at room temperature. Slices were incubated with primary antibodies for 48 hours at 4°C. Afterward, they were washed six times for 10 minutes each in PBS, followed by incubation with fluorophore-conjugated secondary antibodies and DAPI for 2 hours at room temperature. Finally, slices were washed an additional six times for 10 minutes in PBS, mounted together with their membrane inserts onto glass slides, and coverslipped using Fluoromount-G.

### In vitro fibril-induced Tau spreading assay
Hippocampal slices were transduced with AAV viral vector expressing P301S human Tau (AAV2-smCBA-human_P301S_Tau-WPRE; 2×10$^{10}$/mL)

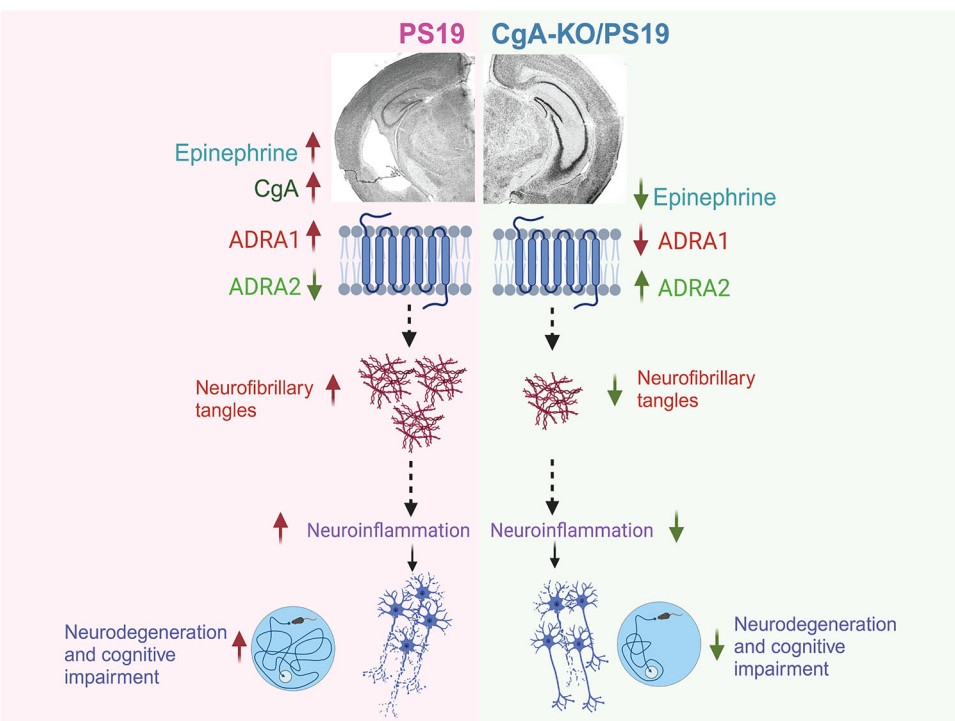

**Fig. 8 | Schematic depicting the role of CgA in Tauopathy.** In Tauopathy brains, elevated levels of chromogranin A (CgA) contribute to increased epinephrine (Epi), accompanied by an upregulation of α1-adrenergic receptors and a downregulation of α2-adrenergic receptors. This imbalance in adrenergic signaling promotes the accumulation of pathological Tau, exacerbates neuroinflammation and neurodegeneration, and accelerates cognitive decline. Genetic depletion of CgA in PS19 Tauopathy mice restores the balance in the adrenergic axis, thereby mitigating Tau-related neuropathology and rescuing cognitive function. (Created in BioRender[85]).

for 24 h. AAV was removed from culture media and K18 (preformed Tau fibrils; 1.5 μg/mL) were added before incubation for 72 h. Subsequently, fresh media without or with Phenylephrine:10 nM, Prazosin:0.5μM, or Epinephrine: 10 nM, was applied as indicated. Slices were harvested on day 21 for Western blot analysis and immunohistochemistry.

### In vivo fibril-induced Tau spreading assay
Mice were anesthetized with inhalation of 2% isoflurane for the duration of surgery and secured on a stereotaxic frame (Kopf Instruments). 3-month-old PS19 and CgA-KO/PS19 mice were injected stereotaxically at a rate of 0.5 μl/min, with 2 μL of 5 mg/ml K18 PFF into the CA1 region of the left hippocampus. The coordinates for injection were anterior-posterior − 2.5, medial-lateral + 2.0, dorsal-ventral −1.8. Mice were sacrificed 6 weeks after injection and transcardially perfused with PBS for immunohistochemistry analysis of Tau aggregation (MC1 + ) in the ipsilateral and contralateral sides.

### K18 purification and in vitro Fibril formation
The repeat domains (K18) of P301L mutant Tau with a Myc Tag was expressed in *E. coli* BL21 (DE3) strain. NaCl (500 mM) and betaine (10 mM), a small molecule chaperone, were added before induction with IPTG (200 μM) at 30 °C for 3.5 h. Cells were resuspended in lysis buffer (20 mM MES, pH 6.8, 1 mM EGTA, 0.2 mM $MgCl_2$, 1 mM PMSF, 5 mM DTT), and passed through a microfluidizer for lysis. After lysis, 5 M NaCl was added to a final concentration of 500 mM followed by heating at 90 °C for 20 min. was clarified by centrifugation and was dialyzed overnight (Buffer A: 50 mM NaCl, 1 mM $MgCl_2$, 0.1 mM PMSF, 2 mM DTT, 1 mM EGTA, 20 mM MES pH 6,8) to obtain the Tau-containing supernatant. Subsequently, cation exchange (HiTrap SP HP, 5 ml column from Cytiva) was performed to purify K18. The degradation products and other impurities were removed by running 15% elution buffer (50 mM NaCl, 1 mM $MgCl_2$, 0.1 mM PMSF, 2 mM DTT,

1 mM EGTA, 20 mM MES pH 6,8, 1 M NaCl). Subsequently, 15-60% gradient (NaCl gradient from 50 mM to 1 M) of NaCl was used to obtain pure K18, which was concentrated and further dialyzed against assay buffer (PBS pH 7.4, 1 mM DTT and 2 mM $MgCl_2$). A 10μM concentration of Tau was incubated in assay buffer for 36 h at 37 °C) upon adding 44μg/ml Heparin. The aggregates formed were snap frozen at −80 °C till further use.

### Transmission electron microscopy
Deeply anesthetized mice were flushed with a pre-warmed (37 °C) Hank's balanced salt solution (HBSS), with calcium and magnesium by cardiac perfusion with freshly prepared pre-warmed (37 °C) fixative containing 2.5% glutaraldehyde and 2% paraformaldehyde in 0.15 M cacodylate buffer using a peristaltic pump. The hippocampus and cortex were dissected and postfixed in 1% $OsO_4$ in 0.1 M cacodylate buffer. The tissues were stained *en bloc* with 2-3% uranyl acetate and dehydrated in graded ethanol series. Sections of 50-60 nm thickness were cut, stained with 2% uranyl acetate and Sato's lead stain. Grids were imaged with a JEOL JEM1400-plus TEM (JEOL, Peabody, MA) attached to a Gatan OneView digital camera with 4k x 4k resolution (Gatan, Pleasanton, CA).

### Measurement of catecholamines
Cortical and plasma catecholamines were measured upon separation by an Atlantis dC18 column (100 A, 3 μm, 3 mm×100 mm) on an ACQUITY UPLC H-Class System attached to an electrochemical detector (ECD model 2465; Waters Corp, Milford, MA) as described previously[18,79]. The mobile phase (isocratic: 0.3 ml/min) contained 95:5 (vol/vol) of phosphate-citrate buffer and acetonitrile. An internal standard 3.4-dihydroxybenzylamine (DHBA 400 ng) was added to the cortical homogenate in 0.1 N HCl. The ECD was set at 500 pA for determination of brain catecholamines. For determination of CSF catecholamines, DHBA (2 ng) was added to 150 μl CSF and adsorbed

with ~15 mg of activated aluminum oxide for 10 min in a rotating shaker. After washing with 1 ml water, adsorbed catecholamines were eluted with 100 µl of 0.1 N HCl. The ECD was set at 500 pA for determination of CSF catecholamines. The data were analyzed using Empower software (Waters Corp, Milford, MA). Catecholamine levels were assessed using the internal standard. Catecholamine levels were provided in nM (CSF) or ng/mg protein. For determination of plasma catecholamines, DHBA (2 ng) was added to 150 µl plasma and adsorbed with ~15 mg of activated aluminum oxide for 10 min in a rotating shaker. After washing with 1 ml water adsorbed catecholamines were eluted with 100 µl of 0.1 N HCl. The ECD was set at 500 pA for determination of plasma catecholamines. The data were analyzed using Empower software (Waters Corp, Milford, MA). Catecholamine levels were normalized with the recovery of the internal standard. Tissue and plasma catecholamines were expressed as ng/mg and ng/ml, respectively. Patient sample details are mentioned in Supplementary Table 1 and Supplementary Table 2.

## Measurement of cytokines

Cortices from PS19, CgA-KO/PS19, WT and CgA-KO were homogenized in PBS followed by centrifugation at 12500 RPM for 30 min at 4 °C. The supernatant was collected and used in ELISA. Cytokines were measured using U-PLEX mouse cytokine assay kit (Meso Scale Diagnostics, Rockville, MD) following the manufacturer's protocol in a MESO SECTOR S 600MM Ultra-Sensitive Plate Imager. The levels were presented in pg/mg protein. Plasma (25 µl) cytokines were measured using U-PLEX mouse cytokine assay kit (Meso Scale Diagnostics, Rockville, MD).

## Behavioral Studies

Behavioral studies including, Morris Water Maze, Novel Object Recognition and Rotarod test for all four cohorts of mice were performed in UCSD behavior core facility.

**Morris Water Maze (MWM) test.** MWM test is a long-term memory test where a chamber is filled with opaque water and there is a submerged platform under the surface. Training was carried out in the period of day1 to day7 for the hidden platform where mice were guided to reach the platform and trial was ended if they reached and are stationary on the platform for 5 seconds. Both distance and time were recorded for each mouse. On day8, the mice were dropped 180 °C opposite of the location of the platform but without the platform in the chamber. Animals were allowed to swim for a maximum of 40 sec before being guided to the platform by the experimentalist. The number of times each mice entering the correct zone was recorded. Separately, the hidden platform was provided, and the time taken by each mouse to reach the platform was recorded and represented as latency.

**Novel object recognition (NOR) test.** In NOR, the mice was left to spend with time with a known object for 180 seconds on day1. The time for which the mouse explored the unknown object was recorded. On day2, a new object is introduced in addition to the old object. It is expected that the mouse will spend more time with the new introduced object for curiosity, if it can remember the old object. The total time spend by mice with the new object on day2 determines the degree of memory loss and represented as (Time with new object X 100)/Time with new + old object.

**Rotarod test.** Rotarod test was performed to evaluate the motor functions in mice. In this test there was a horizontal rod which rotates around its axis and the mice must co-ordinate with the movement so that they don't fall off. The time for which they can stay on the rod determines their motor function. On day 1 each mice undergo five trials on the rotarod. On day 2, each mice undergo seven trials and time from each trial is plotted and compared.

## Tau seeding assay

Human Tau RD P301S FRET biosensor expressing HEK 293 cells (ATCC CRL-3275) were plated on Millipore EZ chambered slide at a confluency of 1 ×103 Cells / well in DMEM Complete media (DMEM with 10% FBS + 100 µg penicillin and streptomycin). After 16 h, the media is replaced with OptiMEM and cells were treated with 2 µg lysate from CgA-KO/PS19 and PS19 cortex prepared in PBS and 4 µl Lipofectamine-3000. Post 24 h of treatment the media is replaced with complete DMEM. After 24 h the media was removed, and cells were fixed with 4% Paraformaldehyde and stained with DAPI. Cells were mounted and images were taken using Keyence Fluorescence Microscope under 40X lens. Four biological replicates per group were taken and 4 images each group were analyzed by ImageJ for quantification.

## Immunohistochemistry

Isoflurane was used to anesthetize the mice followed by trans-cardial perfusion with PBS. The brain was dissected out and kept in Zn-Formalin for 48 hr. After that, Zn-Formalin is replaced with 30% sucrose incubated for 72 hr at 4 °C. The brains underwent coronal sectioning of 30 µm thick by sliding freezing microtome (Epredia). The sections were kept at −20 °C in cryoprotectant. Approximately 7-8 sections were taken for each animal covering from anterior to posterior region of hippocampus. The sections were washed thoroughly (6×10 min wash) with 1X PBS and then incubated with primary antibody in PBS containing 0.4% Triton-X for 24 hr at 4 °C. Following which the primary antibody is removed and washed with 1X PBS for 3X 15 min. Subsequently, the secondary antibody and DAPI (1:2000) were added and incubated for 1 hr at room temperature followed by 3X 15 min PBS wash. Next, the sections were mounted on glass slides with Fluoromount G. The stained slides were imaged under Keyence Fluorescence Microscope. Antibodies used are as follows: AT8 (1:500, Thermo Fisher Science Cat# MN1020), MC1 (1:500, Gift from P. Davies), CD68 (1:700, BIO-RAD Cat# MCA1957GA), GFAP (1:500, Cell Signaling Technology Cat# 3670), ADRA1B (1:500, Abcam, Cat# AB 169523), CgA (1:400, Invitrogen Cat# MA5-13093), NeuN (1:500, Proteintech 66836-1-Ig), Anti-mouse Alexa Fluor 568 (1:300, Invitrogen Cat# A11031), Anti-rabbit Alexa Fluor 488 (1:300, Invitrogen Cat# A10042), Strep-Alexa Fluor 568 (1:500, Thermo Fisher Science Cat#S11226), Strep-Alexa Fluor 647 (1:500, Thermo Fisher Science Cat#S32357).

Paraffin-embedded brain slices pre-mounted on slides from ADRC were de-paraffinized in xylene for 10 mins, followed by rehydration with sequential incubation in 100%, 90%, 70% EtOH, 5 mins each. Sections were placed in Citrate buffer (pH 6) solution in a pressure cooker at low pressure for 10 minutes for antigen retrieval. The slides were blocked (2.5% Normal Goat Serum for 1 hr at room temperature) and incubated with primary antibody for overnight at 4 °C, followed by 3X PBS wash for 15 min each. Next, a secondary antibody and DAPI (1:2000, Thermo Fisher Cat #D3571) were added and incubated for an hour at room temperature. After this, slides were dried and mounted with Fluoromount G. Images were taken with the Keyence Fluorescence microscope.

## Nissl Staining, Volumetric analysis and DG, CA1, and CA3 thickness measurement

Investigators were blinded to the genotypes or treatment of the mice. For quantification of hippocampal volume, mice hemibrains were cut at 30 µm coronally, and all hippocampi, including sections, were collected. Brain sections were mounted on microscope slides (Fisher Scientific) in an anterior-to-posterior order, starting from the section where the hippocampal structure first becomes visible (first section) to the section where hippocampal structure just disappears (last section). Mice with missing sections were excluded from the analyses, a pre-established criterion. Mounted brain sections were dried at room temperature for 24 h and stained with cresyl violet (Nissl staining).

After rehydrating with a quick wash in distilled water, sections were stained in FD Cresyl Violet Solution for 3 min. Next, sections were dehydrated in increasing ethanol concentrations and differentiated in 95% ethanol with 0.1% glacial acetic acid. After the final dehydration in 100% ethanol, sections were cleared in xylene and mounted with DePeX mounting media (VWR). Images were acquired with a Keyence BZ-9000 microscope. Hippocampal volume was estimated using ImageJ (NIH) Volumest plugin (http://lepo.it.da.ut.ee/~markkom/volumest/). To measure the thickness of CA1 pyramidal cell layer and dentate gyrus granule cells layer, a straight line perpendicular to the length of cell layers at fixed locations was drawn and measured using ImageJ.10–12 hippocampal-containing sections were typically used for each analysis.

## Western blotting
15 mg tissue was incubated in RIPA buffer (150 mM NaCl, 1% Triton-X, 12 mM Na-deoxycholate, 0.1% SDS, 50 mM Tris pH 8.0, 25 mM EDTA, 1 mM PMSF, 5% Glycerol, 50 mM NaF, 1 mM $Na_3VO_4$, 1 mM $Na_4P_2O_7$, 25 mM β-Glycerophosphate, 50 mM DTT and PIC) for 10 min. Homogenization was done by handheld motor homogenizer, followed by centrifugation at 12500 rpm for 30 min at 4 °C. The supernatant was collected, and protein concentration was measured using Bradford Protein Assay Reagent. 15 μg protein was loaded in each lane of 12% Tris-Glycine SDS Gel and post-running protein was transferred to PVDF membrane followed by blocking with 5% BSA for 2 hrs. Primary antibody incubation was done at 4 °C. Antibodies and dilution used as follows: AT8 (1:3000, Thermo Fisher Science Cat# MN1020), PHF1 (1:4000, Gift from P. Davies), CP13 (1:2000, Gift from P. Davies), HT7 (1:8000, Thermo Fisher Science Cat# MN1000), Actin (1:8000, BioBharati, Cat# BB-AB0024), ADRA1B (1:2000, Abcam, Cat# AB 169523), ADRA2B (1:2000, Proteintech, Cat# 19778-1-AP), ADRB2 (1:2000, Cell Signaling Technology, Cat# 8513S), Alpha-Tubulin (1:6000, Cell Signaling Technology, Cat# 3873), CgA (1:2000, Invitrogen Cat# MA5-13093), Anti-Mouse-HRP (1:10000, Sigma, Cat# A0168), Anti-Rabbit-HRP (1:10000, Sigma, Cat# A0545), Strep-HRP (1:6000, Thermo Fisher Science Cat#N100).

## cAMP measurement
The cortex tissue samples from mice or human were lysed in PBS along with PDE inhibitor (IBMX) and PIC (SIGMA) with homogenizer followed by centrifugation at 12500 rpm for 20 min at 4 °C. The cAMP level in the supernatant was analyzed using Perkin Elmer cAMP measurement kit as described in the manual. Briefly, the tissue lysate was incubated with anti-cAMP acceptor beads in corning dark 384 well plate and incubated for 30 min at room temperature in dark. Subsequently, biotinylated cAMP/streptavidin beads were added, and incubated at room temperature in dark for 1 h. Excitation was performed at 680 nm wave-length light and emission was measured at 550 nM to determine the cAMP concentration based on the standard curve.

## In vivo Prazosin treatment
The prazosin treatment was administered with utmost care in 6-month-old PS19 mice at a 2 mg/kg body weight dose[80]. A stock concentration of Prazosin was made at 10 mg/ml in DMSO and stored at −20 °C. At the time of injection, the solution was warmed up to 37 °C and diluted in saline containing 10% DMSO. Saline containing 10% DMSO was used for the vehicle-treated experimental set. The mice received treatment twice a week for 12 consecutive weeks, after which they were sacrificed. Half of their brain was frozen for biochemical assays (western blotting) and the other half was fixed with Zn-formalin for immunohistochemistry.

## Cortical Neuronal Culture
Primary neuronal cultures were established from cortices of mouse pups on postnatal day 0 or 1. The cerebral cortices of 3-4 pups were dissected on ice, meninges removed and placed in the cold dissection medium (DM) consisting of 6 mM $MgCl_2$, 0.25 mM $CaCl_2$, 10 mM HEPEs (100X), 0.9% Glucose, 20 mM D-AP5 (Cayman, NC1368401), and 5 mM NBQX (Tocris Bioscience, 10-441-0). The tissues were digested with papain (Worthington, LK003176) in 37 C water bath for 20 min, followed by low OVO and DNase I incubation to stop the digestion for 5 min. The digested tissues were triturated and filtered through 70 mm cell strainer. The cell suspension was centrifuged at 1000 rpm for 10 min, and the cell pellet was gently resuspended in 20 mL B27/NBM/High glucose media and centrifuged again at 850 rpm for 5 min. The resulting cell pellet was resuspended in B27/NBM (1 mL/mouse brain). The cells were counted and plated onto the coverslips at 250,000 in 24-well plates for immunofluorescent imaging. Antibodies used: MAP2 (1:400, Millipore-Sigma Cat#AB-5622), MC1 (1:500, Gift from P. Davies).

## In vitro fibril-induced Tau spreading model
The in vitro fibril-induced Tau spreading model was adapted and modified from previously described[48]. Briefly, primary neurons cultured on coverslips were infected with AAV-P301S PS19 (Virovek, Inc.) on DIV3, and synthetic Tau fibrils (K18/P301L, 100 nM) were added to the medium on DIV5. Neurons without AAV infection (AAV-) and neurons treated with PBS instead of fibrils were included as negative controls. Neurons were incubated with the fibrils for 7–10 days. To test the effect of CgA treatment, recombinant CgA (5 ng/ml; R, D 10422-CH) were added to the culture medium together with the fibrils and replenished once after 3 days. Immunofluorescence staining using the MC1 antibody (1:500) was performed to detect misfolded Tau species. Briefly, neurons grown on coverslips were washed with fresh medium containing 0.01% trypsin, and fixed in fresh 4% PFA with 0.1% Triton-X, followed by regular immunofluorescence staining procedure.

## RNA extraction and qPCR
RNA was isolated using TRIzol reagent and extracted further with phenol-chloroform method. RNA concentration was measured using Thermo Nanodrop and cDNA was synthesized from 500 ng of total RNA using Maxima RT cDNA synthesis kit. SYBR green containing NEB Luna qPCR mix was used to set up the qPCR in Bio-Rad qPCR machine (Primer details mentioned in Supplementary Table 3).

## RNA-seq library preparation and analysis
Total RNA was isolated from prefrontal cortex of 8-month-old mouse using RNeasy kit (Qiagen). RNA amount was quantified by 'Nanodrop' spectrophotometer and integrity was assessed by TapeStation (Agilent). Complementary DNA libraries were prepared from 500 ng of total RNA using the mRNA HyperPrep Kit (KAPA) according to manufacturer's recommendations with Unique Dual-Indexed adapters (KAPA). Libraries were PCR-amplified for 10 cycles and quality was assessed by Tapestation. Libraries were then quantified by Qubit 2.0 fluorometer (Thermo), pooled, and analyzed by paired-end 100 sequencing on the NovaSeq 6000 platform (Illumina) at UCSD IGM Core.

## RNA-Seq Analysis
RNA samples were first assessed for Sequencing quality by FASTQC and libraries with high quality reads were trimmed using Trimmomatic (v0.39). Reads were mapped to the mm10 mouse reference genome using STAR (v2.7.10b) and transcripts were quantified by StringTie (v1.3.6). Transcript counts were imported to R using prepDE.py3 for normalization and differential expression was analyzed by DESeq2 (v1.42.0). Gene clusters among differentially expressed genes were identified as common genes in comparisons of WT vs PS19 and CgA-KO/PS19 vs PS19 with log2 fold-change > +/− 0.5 and p value < 0.01. For gene ontology analysis, the enrichGO function from clusterProfiler (v4.10.0) (Yu et al., 2012) with the org.Mm.eg.db (v3.18.0) was used.

Results were annotated along ontology of biological processes with the following parameters: p-value Cutoff = 0.05, q-value Cutoff = 0.05, and pAdjustMethod = 'BH' (Benjamani and Hochberg).

### Human Transcriptomics Data Analysis

For human transcript analysis, normalized data of gene expression in parahippocampal gyrus (Brodmann area 36) was obtained from Mount Sinai Brain Bank (MSBB) through the Synapse platform (syn16795937). Total number of samples for each Braak stage considered for analysis is mentioned in Supplementary Table 4.

### Statistics

In figure legends 'n' represents the biological replicates for the group. In all in vivo experiments the male and female ratio of 50% was maintained. Normality was tested by Shapiro-Wilk test using GraphPad Prism version 10.4.1. Data were analyzed using unpaired two-tailed T-test, One-way or Two-way ANOVA as mentioned in the figure legends of each figure. Exact p-values are mentioned in the figures. Data are presented as mean values +/− SEM. In all figures except S-Fig4, n represents the number of biological replicates in the experiment.

### Reporting summary

Further information on research design is available in the Nature Portfolio Reporting Summary linked to this article.

## Data availability

The bulk RNA seq data has been generated in this study and deposited at GEO under accession code GSE274459. Source data for all the figures are provided in Source Data File. Source data are provided with this paper.

## Code availability

All the codes used for generating the figures are deposited in Code Ocean (Code Ocean https://doi.org/10.24433/CO.7273160.v1).

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

## Acknowledgements

This work was supported by grants AG072487 to GG, SKM, and XC; AG078635 to SKM and GG; AI163327, GM085490 to GG; R01AG078185, R01AG074273, UCSD Neurosciences start-up funds to XC. IL receives the following NIH grants: 5U01NS112010/807745, U01NS100610, R25NS098999; U19 AG063911-1 and 1R21NS114764-01A1. LT also receives supports from the Michael J Fox Foundation, Parkinson's Foundation, Roche, AbbVie, Lundbeck, EIP-Pharma, Alterity, Novartis, and UCB. SJ was supported by an AFTD Holloway Postdoctoral Fellowship (2022-0002), and DM-M was supported by the AARF-D fellowship from the Alzheimer's Association. The results published here are in whole or in part based on data obtained from the AD Knowledge Portal (adknowledgeportal.org). These data were generated from postmortem brain tissue collected through the Mount Sinai VA Medical Center Brain Bank and were provided by Dr. Eric Schadt from the Icahn School of Medicine at Mount Sinai. We thank UCSD Shiley-Marcos Alzheimer's Disease Research Center (ADRC) for providing AD postmortem brain tissues, blood and CSF samples, and Dr. Dennis Dickson for providing CBD postmortem brain tissue samples from the Jacksonville Mayo Clinic Brain Bank. This work includes data generated at the UC San Diego IGM Genomics Center utilizing an Illumina X Plus that was purchased with funding from a National Institutes of Health SIG grant (#S10 OD026929). We want to acknowledge **UCSD** School of Medicine **Microscopy Core** (Grant P30 NS047101) and **Neuropathology Core** (5P30AG062429).

## Author contributions

Conceptualization: S.K.M., G.G., X.C., S.J. Methodology: S.J., S.K.M., D.D., D.M.-M., S.S., K.T., Y.T. Investigation: S.J., S.K.M., S.S., K.T., D.M.-M., and Y.T. Visualization: S.J., S.K.M., S.S., K.T., D.M.-M., and Y.T. Funding acquisition: S.K.M., G.G., X.C., S.J., and D.M.-M. Project administration: S.K.M., G.G., X.C. Supervision: S.J., S.K.M., G.G., and X.C. Writing – original draft: G.G., S.K.M., X.C., I.L., S.J. Writing – revision: S.K.M., G.G., X.C., and S.J.

## Competing interests

S.K.M. is the founder of CgA Therapeuticals, Inc. GG and SKM are the founders of Siraj Therapeutics. I.L. is a member of the Scientific Advisory Board for the Rossy PSP Program at the University of Toronto, Aprinoia, Amydis, and the Food and Drug Administration (FDA) Peripheral and Central Nervous System Drugs Advisory Committee. She receives her salary from the University of California, San Diego, and as Chief Editor of *Frontiers in Neurology*. The remaining authors declare no competing interests.
