## [Transparent Peer Review file · Nature Communications]

Chromogranin A Deficiency Attenuates Tauopathy by Altering Epinephrine–Alpha-Adrenergic Receptor Signaling in PS19 mice

Corresponding Author: Professor Sushil Mahata

Version 0:

Reviewer comments:

Reviewer #1

(Remarks to the Author)

In this manuscript by Jati et. al. entitled, "Chromogranin A Deficiency Attenuates Tauopathy by Altering Epinephrine–Alpha-Adrenergic Receptor Signaling" authors using patient samples (AD and CBD) and the hTau P301S mouse model, shows that Chromogranin A (CgA) increases Tau pathology, which can be rescued by CgA knockout. The authors also perform a behavioral rescue (MWM test) in KO mice. authors propose that this happens via activation of the epinephrine-alpha-adrenergic receptor (Adra1) pathway by CgA. While this is novel, the mechanistic part feels weak, particularly regarding how epinephrine regulates Tau aggregation. It has been previously shown that CgA colocalize in AD plaques. Also, author did metabolite study, but results are too generalized and not explored in detail in connection with Tau pathology or its role in epinephrine signaling activation. Overall, the experiments are well-designed and executed, and the study is clearly presented. Below are some comments and suggestions that could further improve the manuscript.

- The authors show that glial cells are activated by CgA. Is the pathology observed only in the vicinity of glial cells? Are Tau fibrils bound to CgA, released by microglia taken up by neuronal cells?
- CgA is secreted by neurons. Have the authors checked if CgA increases neuronal uptake of Tau seeds or facilitates cell-to-cell transmission of Tau seeds?
- Calcium influx is altered in Tau pathogenesis, and CgA/Adra1 is linked to calcium flux. Have the authors checked if calcium influx in Tau pathogenesis is affected by CgA? It would be useful to demonstrate this using primary cortical neurons.
- The authors propose signaling axis involved in Tau pathogenesis (CgA-Adra1b/adra2-metabolite? -EPI-Tau pathogenesis). Including a proposed model would help reader understanding.
- In Figure 1, the authors compare Brk 1 with Brk 6 to show elevated levels of CgA and Tau phosphorylation. Did the authors observe any changes between healthy controls and Brk1?
- Do CgA-Tau seeds isolated from patients exhibit increased Tau seeding activity?
- Did author check if colocalization of Adra1b or EPI with CgA and phosphorylated Tau? Are they part of the same complex?
- Did the authors check if the increase in CgA levels in patient samples is due to disease state and not age?
- For Figure 2a, it seems the representative brain slices are not the same. Can author check it? Providing whole brain images/comparing them to the mouse brain ATLAS would be ideal.
- Most immunoblots with CgA KO or Adra do not show immunoblot with CgA protein levels. Including validation of the knockout would be helpful.
- In Figure 3GH, was the same quantity of seed used? Is the lower seeding activity due to lower seeding material?
- CgA is expressed in microglia and neurons. Is the glial activation induced by glial CgA solely responsible for the observed effect? Does neuron secreted CgA induce glial activation?
- The authors propose that EPI produced in C1 neurons of TauP301S mice is released to postsynaptic noradrenergic neurons and drives the observed effect. If so, what is the role of the direct association of CgA with pathogenic Tau?
- Supplementary figures are not numbered or labeled, and it is difficult to find figure legends for supplementary data, making it hard to review.

(Remarks on code availability)

Reviewer #2

(Remarks to the Author)

The manuscript by Jati et al. assesses the role of Chromogranin A (CgA) on tauopathies, specifically Alzheimer's Disease (AD) and Corticobasal Degeneration (CBD). CgA is a secretory prohormone found mostly in endocrine cells but also abundantly released by neurons and neuroendocrine cells. Despite being often released with other catecholamines (i.e., norepinephrine and epinephrine) and widely used as a marker of neuroendocrine tumors, the precise function of CgA in the brain remains elusive. The novelty of the study is investigation of the role of CgA on tau pathogenesis and the consequent cellular responses, i.e., neurodegeneration and glial activation, and memory function. Jati et al. establish the relevance of CgA expression and correlation with tau pathology burden (Braak stage) in AD and CBD patient cohorts. They then utilize a tauopathy model (hTau-PS19), which expresses the human MAPT gene with the P301S mutation commonly found in autosomal dominant tauopathies. The authors show that genetic deletion of Chga (CgA gene) in hTau/ reverses many of the deficits observed in hTau mice, including pathological tau phosphorylation and aggregation, gliosis, spatial learning and memory deficits, and adrenergic signaling. Experiments were conducted using a combination of in vivo and in vitro organotypic slice culture methods. The novel concept and thoroughness of the manuscript enhance our understanding of the protective effect of CgA KO on tau pathogenesis. Major and minor concerns are reported below. Overall, the results are interesting but merit further investigation to establish the importance of the findings for tauopathies and publication in Nature Communications.

Specific points

Point 1: The paper is novel given most reports from the 90s only identified CgA as a component of amyloid plaques and neuritic plaque lesions, but there are no followup reports investigating the direct role of CgA and its downstream pathways in tauopathies. The authors conclude that CgA explains why metabolic conditions such as hypoglycemia, hypotension, and hypoxia are risk factors for neurodegenerative diseases. However, they do not present data to measure any aspects of the metabolic health of their mice, and the metabolites that are measured are not further investigated. Furthermore, Fig. 3B shows no differences in body weight across the genotypes. Thus, it is unclear whether the putative effect of CgA and metabolic syndromes has any connection to the role presented on tau pathogenesis. The authors should present more data to support that the metabolic state of hTau mice is changed/affected by genetic deletion of CgA-KO or revise the abstract, introduction, and results to reduce the focus on the metabolic aspect and confine it to a discussion point later in the manuscript.

Point 2: It is unclear that the CgA antibody used in the paper is specific and has been tested against the appropriate controls. Due to the crucial importance of data presented in Fig. 1/Extended Fig. 1 on the levels of CgA in AD, CBD, and hTau mice, the authors should either provide the source of the CgA antibody and appropriate citations showing that the antibody has been validated elsewhere or show in house validation using their CgA KO mice tissue to prove the specificity of the antibodies.

Point 3: There is no citation or mention of the generation of their CgA KO mice. The authors need to either provide more information on how the mice were generated and characterized or give the readers appropriate citations for previous studies using the same strain. This is important because at this point it is not clear (1) what strategy was used for CgA KO and possible off-target effects; (2) whether the CgA-KO is congenital or inducible; (3) whether the CgA-KO is a global or organ-targeted; (4) what are the pre-determined metabolic conditions altered in the CgA-KO mice used; (5) whether there is any compensation by other chromogranin subtypes (i.e., B and C) in the model. Authors need to address these major points to improve the validity of the model.

Point 4: The results in Fig. 2I-K claim that what the authors observe tau spreading in vivo, but the data does not provide enough evidence to support that claim. The K18 fibrils injected are not fluorescently labeled, and the MC1 antibody would recognize both endogenous and exogenous tau. Additionally, there is no saline-injected control to see if just the injection is enough to cause an increase in MC1+ tau in the ipsilateral vs. contralateral hemisphere. These controls are required to assess tau spreading. Point 5: Even though the authors make great observations regarding the connection between CgA and the adrenergic pathway, the data presented did not fully show a direct relationship between CgA-Adra1-EPI and tau. For instance, in Fig. 5, it is understandable why they chose to treat the WT with the antagonist, and the CgA-KO with the agonist. However, the authors need to treat CgA-KO and WT mice with both the agonist and the antagonist to truly show that the changes in tau pathogenesis are CgA-Adra1 mediated. It would also be important to show that replenishing CgA levels reverses the effects on Adra1 levels/activation and tau pathogenesis. Lastly, experiments performed in Fig. 6F-J, EPI treatment needs to be conducted in CgA-KO organotypic slices to prove that improvements in CgA-KO mice are, in fact, mediated by epinephrine signaling.

Additional points

1. Please define the CgA acronym in the introduction, not only in the abstract.
2. Please expand the introduction to provide a more background on the function of CgA in the periphery and brain. Similarly, an introduction to the adrenergic system and its relationship with tau/CgA should be provided. Currently, the scarcity of information provided hinders the appreciation of the work performed and its conclusions/implications.
3. Please clarify the rationale for choosing the PS19 hTau model. This is a suitable tauopathy model for CBD. However, most of the authors interpretations are tailored to AD, in which the MAPT mutation is not found.
4. Fig. 1D-E – What cell population expresses CgA in the hippocampus? This is a missed opportunity to understand CgA biology. Why are the Western blots (Fig. 1A-B) performed on samples from the frontal cortex and the immunostaining

perform on hippocampal sections? Four samples per group is insufficient to draw significant conclusions. In Fig. 1C, what region of the brain were these data obtained from?

5. Extended data Fig. 1F – There is overlap in Braak I because there is no tau pathology in Braak I. MC1 is a conformational tau antibody. Immunostaining with MC1 would only be apparent at later Braak stage. The immunostaining should be repeated with additional controls and a clear demonstration of the specificity of the CgA antibody (as explained above). The almost complete overlap of co-immunostaining with MC1 raises some doubt about the immunostaining procedure and the antibody specificity.
6. Extended Data Fig. 1C-D - It is unclear why CgA levels were quantified in the CA3 region of hippocampus. Is this the only region where CgA was elevated? Please clarify in the text.
7. Extended data Fig. 1K-L – Please use an additional tau aggregation marker, such as Amytracker to convincingly show tau aggregation. Generally, MC1 is a conformational tau antibody and not necessarily a marker of tau aggregation.
8. Extended data Fig. 2A-B – The authors show that CgA levels are higher in the CA3 region in hTau mice (Extended data Fig. 1C-D). Therefore, it is unclear why they measured brain atrophy in the dentate gyrus and CA1 regions. Please measure thickness of the CA3 for relevance to CgA expression.
9. Fig. 2C – It is unclear why tau phosphorylation and PSD95 levels were measured in cortical rather than hippocampal samples. Please clarify or perform Western blot analysis with hippocampal samples. This is essential for consistency and coherence with the data which indicates that CgA levels are higher in the CA3 region of the hippocampus and the EM data (Extended Figure 2J-K) .
10. Extended Figure 2E - PSD95 data should be shown for WT mice as the control and for consistency.
11. Extended Figure 2J-K – The synapse number should be shown for WT mice as the control and for consistency with Extended Figure 2L-M.
12. Extended Fig. 4A-B - The rationale for using CD68 as the one microglia marker and looking at expression in the CA1 alone (rather than also looking at CA3 and the DG as done in previous figures) is not established. Please provide the data or clarify. Please also include a WT group of mice as the control.
13. On Line #177 the statement that the CgA-KO/hTau has lower levels of CD68 is the wrong interpretation given that the actual data (Extended data Fig. 4A-B) measures only area and no other aspects of microglial morphology. To claim level changes, intensity-based approaches are more appropriate.
14. The section on metabolites is under-developed and does not cohesively fit into the rest of the story.
15. Fig. 4K – The control for the specificity of the Adra1b antibody should be included. The staining pattern for Adra1b appears to be very different between the 3 groups of mice. Please provide an explanation for this variability.
16. Extended Fig. 8A-B - The interpretation that upregulation of Adra1 and downregulation of Adra2 contribute to elevated cAMP levels and the downstream effects is highly speculative and not directly supported by data shown in the manuscript. In fact, given that elevated levels of Adra1 signaling may lead to elevate cAMP levels and that reduced levels of Adra2 signaling may lead to reduced levels of cAMP suggests a potential zero net change in cAMP levels in hTau mice.
17. In Line #116-117 the author states a wrong conclusion “These results suggest that CgA promotes tau phosphorylation and aggregation.”. All data presented is from CgA KO mice, and thus, it does not directly conclude that CgA promotes the tau phosphorylation. Similar overstatements are made in line #136-137 regarding pro-neurodegeneration features of CgA. Please correct results descriptions and conclusions accordingly to match the data presented.
18. On Line #135 please clarify what is meant by “irregular” synapses (i.e., shape, size, volume, structural architecture, etc.).
19. Please define what NE (norepinephrine) stands for in Line #283.
20. Please provide open-access IDs for raw data files from bulk RNA-seq and if possible, attach the analyzed LogFC/p-value datasheet to the extended data. This is an important resource for the research community

Minor points

1. Please add to the animal's section the source of the mice used in the study (both the CgA-KO and hTau). Please cite the appropriate papers on how these mice were generated, characterized, and validated. Additionally, I would strongly advise authors to refer to the hTau model, or minimally introduce it, by its widely known strain name PS19.
2. Please include how long (seconds/minutes) the animal was allowed to swim before being guided to the platform in the MWM.
3. The citation on line #411 for organotypic slice cultures is not in the right citation format as the rest of the manuscript.
4. If performed, please add a statement about data normality testing and usage of non-parametric tests (when applicable). If data normality was not performed, please justify.
5. Please include the source (company and catalog #) of antibodies used for both WB and ICC. The dilutions used are provided, but for example, it is unclear which are the CgA and total tau antibodies used and if they have been previously validated.
6. Fig 1C and J should be moved to the supplements as the authors do not pursue differences in the change at the protein or transcript levels of CgA and move forward with a CgA KO model.
7. Fig 1C is cited after figures 1D-E in the results section; Extended figure 1E is cited before 1C and D. Please correct.
8. Relative to extended figure 1H, please address the difference in molecular weight of tau in WT animals seeded with K18 relative to the other experimental conditions.
9. Representative images in Extended Fig 1K and main Fig. 5E make unclear what regions are being assessed in the hippocampal OTSC, which makes the MC1 staining hard to interpret. Please include neuronal marker stain or a better image showing the CA3 region. Please address this in the text.
10. Data in Extended Fig. 3C and D should be moved to the main manuscript Fig. 3 since they add to the behavioral validity of the model.
11. Relative to extended Fig. 2K-M and main Fig 3B - please show individual points for consistency with other graphs throughout the paper.
12. Relative to extended Fig. 4A and B, and main Fig. 4K the CD68 representative image in hTau mice has more background-like signal than the CgA KO. Please clarify if images were processed similarly (i.e., during acquisition and post-

processing). Additionally, if there are discrepancies in the signal specificity, measurement of area in this case may not be the best endpoint and authors should consider looking at either the intensity of CD68 signal or stereological analysis of microglia number/morphology.

13. Please clarify why Extended Fig 5 (addressing pro-inflammatory cortex and plasma cytokines) is separate from Extended Fig. 4 and cited before Extended Fig. 4G-L, perhaps glia data should be consolidated in Extended Fig. 4 and metabolic data – now spread across Extended data Fig. 4I-L and Extended Fig. 6 - should be in a separate figure.

14. Fig. 4C is mentioned before Fig. 4A and B, please correct.

15. Fig. 6C is mentioned before Fig. 6A and B, please correct.

(Remarks on code availability)

Reviewer #3

(Remarks to the Author)

Jati and colleagues present a work entitled 'Chromogranin A Deficiency Attenuates Tauopathy by Altering Epinephrine–Alpha-Adrenergic Receptor Signaling' in which they study the role of chromogranin A changes in human tauopathies and experimental models. First, they found an increased amount of chromogranin A in the autopsy-derived brain materials from patients with Alzheimer's disease or corticobasal degeneration. Such increase was also found in the brains of tau transgenic mice (PS19). They then crossed Chromogranin A knockout mice with PS19 transgenic mice. Neuropathological changes and cognitive functions were preserved in CgA-KO/PS19 mice. Transcriptomic data allowed for the identification of changes in adrenergic receptors expression. Knowing that chromogranin A plays an important role in the adrenergic system. They then studied the alpha adrenergic receptor and epinephrine in tau transgenic mice, focusing on the inflammatory component. Their results suggest that the Alpha adrenergic receptor/epinephrine system plays an important role in the connection between metabolic regulation and neurodegeneration.

These findings are original to the field but also related fields like metabolism.

Altogether, the work is original but the storytelling might be improved. For instance, the ex vivo work on CgA-KO brain slices in extended figure 1 panels G-L is more related to the next section in CgA-KO/PS19 mice.

Regarding the data, some conclusions may be further supported by the addition of controls.

Comments:

In the field of tau research, htau mice refer to a well-described strain developed by the laboratory of Peter Davies and available from The Jackson Lab (<https://www.jax.org/strain/005491>). The strain used in the present is known as TauP301S (line PS19) (<https://www.alzforum.org/research-models/tau-p301s-line-ps19>). Please modify to avoid any confusion.

In patients, there is a mix of CBD and PSP. Neuropathology indicates ARTAG suggesting glial tau positive inclusions. There is no information about CgA immunoreactivity in astrocytes. Do the authors investigate such co-localization?

HT7 is an antibody which recognizes human tau and not mouse tau. Is murine tau modulated in CgAKO mice? What are the levels of phosphorylation in murine tau? In the manuscript, there is no analysis of murine tau phosphorylation and expression in WT and CgAKO. It must be explored to precise what is the connection between Alpha adrenergic receptor/epinephrine and tau phosphorylation. Is it physiological or only there in tau overexpression models? Similarly, many data from CgAKO are missing. For instance in extended Figure 2, nothing on thickness. Conversely, in inflammation studies, the four animal groups are studied. Why some groups are lacking in other experiments? Murine tau, life span...

The use of immunoblotting for Chromogranin A quantification is not satisfactory. Many immunological assays for CgA are available since it is a well-described biomarker in oncology.

Surprisingly, there is no behaviour tests on fear and depression where epinephrine is a key factor. For pharmacological studies with prazosin and phenylephrine, is there any dose-response study?

The figures and extended figures are very busy and difficult to understand.

In figure 1, the panel 1C of CgA mRNA expression is not explicit. The y-axis indicates normalized counts but there is no indication of mRNA (except in the figure legend).

In figure 1D, there is no information about the area (CA or DG) and cell layers (granular cell layer, pyramidale layer, etc...) of the hippocampus where high magnification were taken. There is no scale bar in Figure 2A, and hippocampus of the CgA-KO/hTau (PS19) panel looks bigger.

Very nice work which may be improved

(Remarks on code availability)

Reviewer #4

(Remarks to the Author)

(Remarks on code availability)

Version 1:

Reviewer comments:

Reviewer #1

(Remarks to the Author)

In the revised manuscript, the authors have satisfactorily addressed most of the concerns raised in the original submission. The new data demonstrate that neuronal uptake of Tau seeds is significantly increased in the presence of CgA. Additionally, the manuscript now includes data on the colocalization of adra1B and pTau. The authors have provided validation experiments for the CgA antibody, though the corresponding figure is included only for the reviewer. The metabolic section, which was weaker in the previous version, has been removed as per the suggestion of another reviewer in consultation with the Editor, resolving some of the previously raised issues. Overall, the revisions have strengthened the manuscript, and it is now suitable for publication.

(Remarks on code availability)

Reviewer #2

(Remarks to the Author)

The revised manuscript by Jati et al. has significantly improved following suggestions from all the Reviewers. The authors have improved the flow of the manuscript by removing the incomplete metabolite data and focusing on the tau dysregulation. They have performed an extensive number of new experiments/analyses to address concerns related to CgA antibody specificity, in vivo tau injection experiments, and directionality of the CgA-Adra1-EPI-tau axis. Additionally, they have provided improved data representations and convincing explanations throughout the rebuttal letter when new experiments were not possible. The current manuscript requires a few minor edits prior to publication (detailed below).

The points below specifically refer to the points addressed in the Rebuttal letter.

Point 2: The authors have substantially reduced concern regarding specific detection of CgA in Fig. R2. Please provide a CgA Western blot with higher exposure for Fig. 2C, similar to the robust detection observed in Fig. R2 and the other blots in Fig. 2C.

Additional points

7. We agree with the authors about the nature and description of MC1 binding, and its suitability to detect misfolded tau. However, misfolded tau is not the same as aggregated tau, and using MC1 in IHC analysis does not permit to confidently address tau aggregation. For instance, the paper cited above (PMID: 24271788) does not refer to the MC1 staining as aggregated tau, but as misfolded tau. Meanwhile, the current manuscript extrapolates the MC1+ to tau aggregation in multiple places such as lines #138,159,274,276,298-306,312,339,342,349,473, and 658. Please revise the statements in the text to appropriately address the misfolded species of tau that is labeled with MC1.

Furthermore, the suggestion to include validated markers (e.g., Amytracker), which detect features of tau aggregates (i.e., amyloid structure) was to allow the authors to confidently establish the effects on tau aggregation and not just misfolding. However, the image provided for the Amytracker staining in E-Fig. 1N is not convincing and looks like non-specific signal in absence of appropriate controls (i.e., no dye, dilution curve, more sample size, etc.). We appreciate the effort of the authors to provide this piece of data, but in its current state, it should be removed from the manuscript. The authors should revise the language in the manuscript to "misfolded tau" as discussed above or provide more convincing data for the formation of true tau aggregates that form amyloid structure.

13. This was not addressed and the previous phrase (now on line #213) still incorrectly states decrease in levels of CD68 while only area was measured. Please correct.

20. The point was addressed. However, it is highly recommended that, upon publication, the tables are provided as searchable Excel files instead of tables in PDF files as it is currently presented.

Minor points

1. The point was addressed for the CgA KO mice. However, as pointed also by Reviewer #3, the authors should modify the

hTau nomenclature to PS19 because that is the widely known name of the strain and referring to it as hTau gives the wrong impression that these mice express non-mutated human tau.

2. Please include this information in the methods section of the MWM.

4. This was not addressed nor added to the statistical methods section.

(Remarks on code availability)

Reviewer #3

(Remarks to the Author)

The authors' answer is satisfactory. However, since no immunological tests other than immunoblot were used, it would be desirable to visualise the entire blot in order to identify the fragments derived from chromogranin A.

(Remarks on code availability)

Reviewer #4

(Remarks to the Author)

(Remarks on code availability)

RESPONSE TO REVIEWERS' COMMENTS

We thank all reviewers for their careful reading of the manuscript and their depth of knowledge of the subject area. We also appreciate their positive feedback on our manuscript as reflected by their use of words or phrases such as *'novel, 'well-designed experiments,' and 'clear presentation'* (reviewer 1); *'novel,' 'thorough,' and 'interesting'* (reviewer 2); and *'original' and 'very nice'* (reviewer 3). All reviewers identified some weaknesses in the manuscript. We have carefully gone through each criticism and conducted several new experiments to address their concerns. Below is a summary of changes we made in the manuscript.

Major changes

- We removed the metabolism section, as suggested by Reviewer 2, since we agreed that it required in-depth experimentation beyond the scope of this work.
- Consequently, we also decided **not** to perform any new experiments on calcium signaling.

These changes were permitted by the editor.

In response to reviewers' comments, we performed 13 new experiments:

1. Alzheimer's Disease (AD) Hippocampus immunoblotting (**Fig. 1B&D**).
2. Low magnification images of Braak6 and Braak1 stage AD samples (**E-Fig. 1C**).
3. Braak 0 Hippocampus IHC (**E-Fig. 1J**).
4. Cortical neuronal culture (**Fig. 1K&L**).
5. Amytracker staining (**E-Fig. 1N**).
6. Cortex western blot includes WT and CgA Blot (**Fig. 2C**).
7. Hippocampus western blot (**E-Fig. 2G-J**).
8. Iba1 IHC with hTau, WT and CgA-KO/hTau (**Fig. 4E&F**).
9. CgA immunoblot was done for the adrenergic receptor figure (**Fig. 5H**).
10. IHC for the spatial distribution of CgA, Adra1b, and MC1 (**E-Fig. 6A-D**).
11. *In vivo* Prazosin treatment experiment (**Fig. 6E-H, E-Fig. 8D-F**).
12. Testing effects of epinephrine and prazosin on the involvement of Adra1 in Tauopathy (**Fig. 7K-M; E-Fig. 9M-O**).
13. Dose-dependent effects of PR and PE in CgA-KO and WT OTSC, respectively (**E-Fig. 8D&G**).
14. Testing effects of EPI on CgA-KO OTSC (**E-Fig. 9H-L**).

Point-by-point response to reviewers' comments

Response to Reviewer #1:

In this manuscript by Jati et. al. entitled, "Chromogranin A Deficiency Attenuates Tauopathy by Altering Epinephrine–Alpha-Adrenergic Receptor Signaling" authors using patient samples (AD and CBD) and the hTau P301S mouse model, shows that Chromogranin A (CgA) increases Tau pathology, which can be rescued by CgA knockout. The authors also perform a behavioral rescue (MWM test) in KO mice. authors propose that this happens via activation of the epinephrine-alpha-adrenergic receptor (Adra1) pathway by CgA. While this is novel, the mechanistic part feels weak, particularly regarding how epinephrine regulates Tau aggregation. It has been previously shown that CgA colocalize in AD plaques. Also, author did metabolite study, but results are too generalized and not explored in detail in connection with Tau pathology or its role in epinephrine signaling activation. Overall, the experiments are well-designed and executed, and the study is clearly presented. Below are some comments and suggestions that could further improve the manuscript.

Response: We thank the reviewer for positive evaluation of our manuscript and for insightful comments. We also thank the reviewer for recognizing that our work is *'novel'*, that the *'experiments are well-designed, and executed'* and that the *'results are clearly presented'*.

The authors show that glial cells are activated by CgA. Is the pathology observed only in the vicinity of glial cells? Are Tau fibrils bound to CgA, released by microglia taken up by neuronal cells?

Response: Our study was not specifically aimed at investigating the activation mechanism of glial cells by CgA. Rather, we undertook a holistic approach to assess the outcome of CgA ablation in the development of Tauopathy-mediated neurodegeneration, where microglial activation is curbed and serves as one of the

mechanisms underlying our observations. However, in the cortex of CgA-KO/hTau mice, we observed a rescue of inflammatory markers **CD68**, **Iba1**, and inflammatory cytokines compared to hTau (**Fig. 4**). The pathology was detected in different regions of the hippocampus, including the vicinity of glial cells. Although current evidence does not support the expression of Tau in microglia (PMID: 30227881), the reviewer's suggestion presents an interesting avenue for future investigation.

CgA is secreted by neurons. Have the authors checked if CgA increases neuronal uptake of Tau seeds or facilitates cell-to-cell transmission of Tau seeds?

Response: We appreciate the reviewer for recognizing this interesting issue. Our new data show that neuronal uptake of Tau seeds was significantly increased in the presence of CgA (5 ng/ml) as detected by MC1 staining (please see new **Fig. 1K&L**).

Calcium influx is altered in Tau pathogenesis, and CgA/Adra1 is linked to calcium flux. Have the authors checked if calcium influx in Tau pathogenesis is affected by CgA? It would be useful to demonstrate this using primary cortical neurons.

Response: We thank the reviewer for raising this issue. Altered calcium influx is known to contribute to neurodegeneration in AD/Tauopathy. We decided not to include the metabolic consequences in our study, as this would require extensive experimentation. Further, another reviewer suggested to remove the section with metabolomics data, which we have done with the editor's permission. In a future study, we plan to take a deeper look at calcium flux and metabolic changes in hTau, as well as their recoveries in CgA-KO/hTau.

The authors propose signaling axis involved in Tau pathogenesis (CgA-Adra1b/adra2-metabolite? -EPI-Tau pathogenesis). Including a proposed model would help reader understanding.

Response: As suggested by this reviewer, we have now included our proposed model in a new Fig. 8.

In Figure 1, the authors compare Brk 1 with Brk 6 to show elevated levels of CgA and Tau phosphorylation. Did the authors observe any changes between healthy controls and Brk1?

Response: We have included new data showing no noticeable difference in MC1 and CgA staining of hippocampus between control (Braak 0) and Braak 1 (**see new figure, S-Fig 1. E&H**). Please note that the sample size for Braak 0 is small (n=1). Due to the scarcity of samples, and that Braak 1 typically has minimal tau pathology, therefore were included as part of our control group.

Do CgA-Tau seeds isolated from patients exhibit increased Tau seeding activity?

Response: In this study, we did not isolate CgA-Tau seeds. However, we will evaluate this in a future study.

Did author check if colocalization of Adra1b or EPI with CgA and phosphorylated Tau? Are they part of the same complex?

Response: We agree with the reviewer. We have conducted new experiment to address this issue. Our new data show a near perfect colocalization of Adra1b with Tau in the CA2, CA3 and CA4 regions of the hippocampus. CgA, however, is seen as punctate vesicles in axons and dendrites and colocalized with Tau (MC1 staining) primarily in the cell body (soma) of CA2 and CA3 region (please **see new figure E-Fig. 6**).

Dysregulation of autolysosome and accumulation of proteinopathic aggregates in soma known to serve as source for Tau spreading (PMID: 35654956). Considering this report, our finding of increased CgA protein levels and its associating with Tau fibrils in soma shed some lights into how CgA might be involved in Tau spreading. The association between CgA and Tau seeds (MC1⁺) was observed in the Braak VI patient samples (**E-Fig. 1H**). These results suggest that the effect of CgA on Adra1b, although indirect, drives the progression of Tauopathy through catecholamine-mediated signaling pathways.

- Did the authors check if the increase in CgA levels in patient samples is due to disease state and not age?

Response: Since the patient samples were nearly age-matched (supplementary table), we assume that the elevated levels of CgA reflect an association with the disease state.

Fig. R1. Series of hippocampus image for WT, hTau, CgA-KO, and CgA-KO/hTau. Scale bar: 200 μ m.

• For Figure 2a, it seems the representative brain slices are not the same. Can author check it? Providing whole brain images/comparing them to the mouse brain ATLAS would be ideal.

Response: We confirm that the representative images are from the same positions based on the brain Atlas coordinate (Bregma -2.18 mm, interaural 1.62 mm). Below (**Fig. R1**) is an example of Nissl image series of WT, hTau, CgA-KO and CgA-KO/hTau whole brain.

• Most immunoblots with CgA KO or Adra do not show immunoblot with CgA protein levels. Including validation of the knockout would be helpful.

Response: We totally agree and thankful to this reviewer for raising a valid and important point. We have now validated CgA antibody using pure CgA (5 ng) as a positive control in addition to tissue lysates from mouse cortex (20 μ g), and human frontal cortex (10 μ g). Anti-CgA antibody used throughout our study detects CgA protein where the monomeric form migrates at identical positions (**Fig. R2A**). The same antibody could not detect any protein at the corresponding position in samples prepared from CgA-KO/hTau brain extracts (please see revised **Fig. 2C**). We have also shown increased expression of CgA in 9-month-old hTau, reinforcing augmented expression of CgA in AD and CBD patient samples. In addition, we found decreased expression of Adra1b and no expression of CgA in 9-month-old CgA-KO/hTau cortex, confirming validity of the absence of CgA in CgA-KO mice (see revised **Fig. 4H**). The absence of CgA mRNA in CgA-KO/hTau (**Fig. R2B**) also coincides with lack of CgA protein in CgA-KO/hTau. We also performed CgA IHC in CgA-KO and WT which confirmed the depletion of CgA signal (**Fig. R2C**). Taken together, these results validate authenticity of both the anti-CgA antibody and CgA-knock out mouse.

with lack of CgA protein in CgA-KO/hTau. We also performed CgA IHC in CgA-KO and WT which confirmed the depletion of CgA signal (**Fig. R2C**). Taken together, these results validate authenticity of both the anti-CgA antibody and CgA-knock out mouse.

Fig. R2. A. Western blot with pure protein and lysate from mouse and human extract. B. RNA seq data showing depletion of CgA mRNA in CgA-KO/hTau. C. IHC showing depletion of CgA in CgA-KO/hTau. Scale bar = 50 μ m

In Figure 3GH, was the same quantity of seed used? Is the lower seeding activity due to lower seeding material?

Response: We apologize for the lack of clarification of the experiment. This experiment was to assess the seeding capacity of the same amounts of total protein in the cortical lysates of hTau and CgA-KO/hTau, not the same amount of the seed itself. In this experiment, we homogenized the cortex in PBS and used equal amount of protein from hTau and CgA-KO/hTau cortices for the seeding experiment in the HEK 293-RD biosensor. The lower green fluorescence from the FRET signal in HEK 293-RD cells indicates more seeding activity of the seeds present in CgA-KO/hTau cortical lysates compared to equal amount of WT lysates.

CgA is expressed in microglia and neurons. Is the glial activation induced by glial CgA solely responsible for the observed effect? Does neuron secreted CgA induce glial activation?

Response: Although we agree and recognize this as an important point, we did not investigate the specific role of CgA in glia vs neuron. We will investigate this in future.

• The authors propose that EPI produced in C1 neurons of TauP301S mice is released to postsynaptic noradrenergic neurons and drives the observed effect. If so, what is the role of the direct association of CgA with pathogenic Tau?

Response: While we very much appreciate this reviewer for raising this deep mechanistic question, we plan to address this important issue in future work. We speculate that CgA exerts its pathogenic effects directly by binding to Tau or indirectly via the release of EPI from C1 neurons. We have included this hypothesis in the Discussion.

Supplementary figures are not numbered or labeled, and it is difficult to find figure legends for supplementary data, making it hard to review.

Response: We are extremely sorry for the inadvertent mistakes. We have now clearly labeled supplemental figures and figure legends.

Response to Reviewer #2:

The manuscript by Jati et al. assesses the role of Chromogranin A (CgA) on tauopathies, specifically Alzheimer's Disease (AD) and Corticobasal Degeneration (CBD). CgA is a secretory prohormone found mostly in endocrine cells but also abundantly released by neurons and neuroendocrine cells. Despite being often released with other catecholamines (i.e., norepinephrine and epinephrine) and widely used as a marker of neuroendocrine tumors, the precise function of CgA in the brain remains elusive. The novelty of the study is investigation of the role of CgA on tau pathogenesis and the consequent cellular responses, i.e., neurodegeneration and glial activation, and memory function. Jati et al. establish the relevance of CgA expression and correlation with tau pathology burden (Braak stage) in AD and CBD patient cohorts. They then utilize a tauopathy model (hTau-PS19), which expresses the human MAPT gene with the P301S mutation commonly found in autosomal dominant tauopathies. The authors show that genetic deletion of *Chga* (CgA gene) in hTau/ reverses many of the deficits observed in hTau mice, including pathological tau phosphorylation and aggregation, gliosis, spatial learning and memory deficits, and adrenergic signaling. Experiments were conducted using a combination of in vivo and in vitro organotypic slice culture methods. The novel concept and thoroughness of the manuscript enhance our understanding of the protective effect of CgA KO on tau pathogenesis. Major and minor concerns are reported below. Overall, the results are interesting but merit further investigation to establish the importance of the findings for tauopathies and publication in Nature Communications.

Response: We thank the reviewer for carefully reading the manuscript and recognizing our work as 'novel,' 'thorough,' and 'interesting'.

Specific points:

Point 1: The paper is novel given most reports from the 90s only identified CgA as a component of amyloid plaques and neuritic plaque lesions, but there are no follow up reports investigating the direct role of CgA and its downstream pathways in tauopathies. The authors conclude that CgA explains why metabolic conditions such as hypoglycemia, hypotension, and hypoxia are risk factors for neurodegenerative diseases. However, they do not present data to measure any aspects of the metabolic health of their mice, and the metabolites that are measured are not further investigated. Furthermore, Fig. 3B shows no differences in body weight across the genotypes. Thus, it is unclear whether the putative effect of CgA and metabolic syndromes has any connection to the role presented on tau pathogenesis. The authors should present more data to support that the metabolic state of hTau mice is changed/affected by genetic deletion of CgA-KO or revise the abstract, introduction, and results to reduce the focus on the metabolic aspect and confine it to a discussion point later in the manuscript.

Response: As suggested by the reviewer, we have excluded metabolite data and updated the MS accordingly.

Point 2: It is unclear that the CgA antibody used in the paper is specific and has been tested against the

appropriate controls. Due to the crucial importance of data presented in Fig. 1/Extended Fig. 1 on the levels of CgA in AD, CBD, and hTau mice, the authors should either provide the source of the CgA antibody and appropriate citations showing that the antibody has been validated elsewhere or show in house validation using their CgA KO mice tissue to prove the specificity of the antibodies.

Response: Please see our response above to Reviewer 1 (**Fig. R1A-C**).

Point 3: There is no citation or mention of the generation of their CgA KO mice. The authors need to either provide more information on how the mice were generated and characterized or give the readers appropriate citations for previous studies using the same strain. This is important because at this point it is not clear (1) what strategy was used for CgA KO and possible off-target effects; (2) whether the CgA-KO is congenital or inducible; (3) whether the CgA-KO is a global or organ-targeted; (4) what are the pre-determined metabolic conditions altered in the CgA-KO mice used; (5) whether there is any compensation by other chromogranin subtypes (i.e., B and C) in the model. Authors need to address these major points to improve the validity of the model.

Response: The details of the generation of the CgA-KO mice have been described in our earlier publication (PMID: 16007257). Below are the clarifications to this reviewer's concerns:

(1) *Cre-loxP* gene-targeting strategy was used to generate CgA-KO mice

(2) CgA-KO mice are congenital

(3) CgA-KO is a global knockout

(4) CgA is a unique pro-protein having counter-regulatory motifs such as Pancreastatin (PST: hCgA₂₅₀₋₃₀₁)

domain is pro-diabetic (PMID: 3537810; PMID: 15956083; PMID: 19706599; PMID: 25048197;

PMID: 28228748) and pro-inflammatory (PMID: 28228748) as opposed to Catestatin (CST: hCgA₃₅₂₋₃₇₂)

domain, which exerts anti-diabetic (PMID: 29432123) and anti-inflammatory (PMID: 29432123;

PMID: 29304538; PMID: 33826401) effects. In addition, CST exerts anti-adrenergic (PMID: 9294131;

PMID: 11043569; PMID: 12799369) and anti-hypertensive (PMID: 9786174; PMID: 16007257;

PMID: 23129699; PMID: 24731867; PMID: 33826401) effects. In view of the above, lack of CgA in the CgA-KO

mice became supersensitive to insulin (owing to the lack of PST), hyperadrenergic and hypertensive (owing to

the lack of CST). (5) While one group found no compensatory increase in Chromogranin B and C (aka

secretogranin II) (PMID: 16007257), the other group found compensatory increase in expression of

Chromogranin B (PMID: 18367602) but not Secretogranin II.

Point 4: The results in Fig. 2I-K claim that what the authors observe tau spreading in vivo, but the data does

not provide enough evidence to support that claim. The K18 fibrils injected are not fluorescently labeled, and the MC1 antibody would recognize both endogenous and exogenous tau. Additionally, there is no saline-injected control to see if just the injection is enough to cause an increase in MC1+ tau in the ipsilateral vs. contralateral hemisphere. These controls are required to assess tau spreading.

Fig. R3. A: Saline injected PS19 showing no MC1 staining. B: K18 injected PS19, showing MC1 spreading. C: Myc staining showing K18 injection (K18 is Myc Tagged) in the ipsilateral part, and K18 is not spread to the contralateral hippocampus. D: MC1 staining shows the spreading of Tau from the ipsilateral to the contralateral hippocampus of the same brain in C. Scale bar = 200 μ m.

Response: We agree with the reviewer and thankful for pointing out the lack of these important controls, which we have properly addressed now. First, as suggested by the reviewer, we performed stereotaxic injection of PBS to the same coordinates of the brain, which showed no MC1 staining (**Fig. R3A**). In contrast, injection of K18 fibrils (dissolved in PBS) showed MC1 staining in both the ipsilateral (injection side) and contralateral side (Tau

spread side), with the ipsilateral side having more MC1+ signaling, as expected (**Fig. R3B**). To distinguish the signal coming from the K18 fibril versus endogenous Tau aggregates, we performed immunostaining using anti-Myc antibody, since the injected K18 is Myc tagged. We found that the antibody stained only locally in the injected ipsilateral area (along the needle track and at the targeted coordinates) and not in CA3 or the contralateral side (**Fig. R3C**). In contrast, MC1 staining of the same brain detected MC1+ Tau aggregates on both ipsilateral (CA3) and contralateral sides (**Fig. R3D**), implicating that the MC1+ Tau aggregates observed in the K18 injected brains

largely came from fibril seeding-induced endogenous Tau aggregation and propagation, rather than the injected Myc-K18 fibrils themselves. The K18 Tau fibril injection induced Tau spreading model has been published previously (PMID: 23325240, 31906970, 38657612).

Point 5: Even though the authors make great observations regarding the connection between CgA and the adrenergic pathway, the data presented did not fully show a direct relationship between CgA-Adra1-EPI and tau. For instance, in Fig. 5, it is understandable why they chose to treat the WT with the antagonist, and the CgA-KO with the agonist. However, the authors need to treat CgA-KO and WT mice with both the agonist and the antagonist to truly show that the changes in tau pathogenesis are CgA-Adra1 mediated. It would also be important to show that replenishing CgA levels reverses the effects on Adra1 levels/activation and tau pathogenesis. Lastly, experiments performed in Fig. 6F-J, EPI treatment needs to be conducted in CgA-KO organotypic slices to prove that improvements in CgA-KO mice are, in fact, mediated by epinephrine signaling.

Response: We agree with the reviewer in that we have not established a direct link between CgA and the adrenergic pathway. In WT organotypic slice culture, we found prominent pathogenic tau inclusion staining with MC1 antibody in presence of EPI as compared to the presence of PBS, implicating a direct induction of Tau pathogenesis by EPI. We also found significant decrease in MC1 staining in presence of EPI and Prazosin, indicating EPI-induced Tau pathogenesis is mediated via Adra1 receptor (**see new figure, Fig. 7K-M**). Invivo prazosin treatment was also effective in reducing Tau phosphorylation and aggregation in hTau mice (**Fig 6E-H, E-Fig 8J**). We have also tested the effect of CgA treatment on exacerbating tau pathogenicity. In primary neurons expressing hTau, CgA-treatment induces tau aggregation (**Figure. 1 K-L**). Also the increase in EPI induced Tau phosphorylation and aggregation is not significant in CgA-KO OTSC (**E-Fig H-L**) owing to reduced expression of Adra1. Collectively, these data strongly suggest a direct role of the CgA-Adra axis in tau pathogenicity.

Additional points

1. Please define the CgA acronym in the introduction, not only in the abstract.

Response: We have now provided CgA acronym in the very 1st line in the Introduction (line 56).

2. Please expand the introduction to provide a more background on the function of CgA in the periphery and brain. Similarly, an introduction to the adrenergic system and its relationship with tau/CgA should be provided. Currently, the scarcity of information provided hinders the appreciation of the work performed and its conclusions/implications.

Response: As suggested by the reviewer, we have provided a more background on the function of CgA in the periphery and brain (lines 56-72). In addition, we have introduced the adrenergic system and its relationship with Tau/CgA (lines 83-90).

3. Please clarify the rationale for choosing the PS19 hTau model. This is a suitable tauopathy model for CBD. However, most of the authors interpretations are tailored to AD, in which the MAPT mutation is not found.

Response: Indeed, as the reviewer pointed out, the PS19 model is precisely an FTD/primary tauopathy model. Our intention is to use the PS19 mice to model tauopathy, which is a shared feature across primary tauopathies (e.g. PSP, CBD) and secondary tauopathies including AD. Although we started our study with the observation that CgA was increased and colocalized with tau aggregates in AD brains, we went on to determine the causal relationship between CgA and tau in PS19 mice, followed by mechanistic studies. We have clarified these points in the revised manuscript.

4. **Fig. 1D-E** – What cell population expresses CgA in the hippocampus? This is a missed opportunity to understand CgA biology. Why are the Western blots (Fig. 1A-B) performed on samples from the frontal cortex and the immunostaining performed on hippocampal sections? Four samples per group is insufficient to draw significant conclusions. In Fig. 1C, what region of the brain were these data obtained from?

Response: Using in situ hybridization, Mahata et al. (PMID: 12106456) demonstrated the highest expression of CgA in CA2, CA3 and CA4 region, moderate expression in CA1, granule cell layer of rat hippocampus.

Immunohistochemical studies revealed highest expression of CgA in GABAergic interneurons scattered through all layers of CA2 and CA3 weak expression of CgA in the glutamatergic neurons located in the pyramidal cell layer (PMID: 20005907). Consistent with these findings, our results also showed CgA expression primarily in CA2 and CA3 granular layer (E-Fig. 6). This observation matches with the CgA staining in human AD hippocampus where we observed maximum CgA staining mostly in CA2 and CA3 region (E-Fig 1C)

We also performed western blot from AD hippocampus sample (Fig. 1B&D) and with increased number of samples for hippocampus IHC (Fig. 1F).

5. Extended data Fig. 1F – There is overlap in Braak I because there is no tau pathology in Braak I. MC1 is a conformational tau antibody. Immunostaining with MC1 would only be apparent at later Braak stage. The immunostaining should be repeated with additional controls and a clear demonstration of the specificity of the CgA antibody (as explained above). The almost complete overlap of co-immunostaining with MC1 raises some about the immunostaining procedure and the antibody specificity.

Response: We have now included Braak 0 stage immunostaining. We found scattered CgA staining in these tissues. As described above (in response to reviewer 1), we have validated the antibody using pure CgA protein as a positive control and tissue lysates from mouse and human cortices (Fig. R1). As demonstrated earlier, we found the association of Tau and CgA only in soma in hTau mice (E-Fig. 6) and also in patient samples (E-Fig. 1H). These results underscore the importance of CgA in spread of proteinopathic aggregates which initiates from soma (PMID: 35654956).

6. Extended Data Fig. 1C-D - It is unclear why CgA levels were quantified in the CA3 region of hippocampus. Is this the only region where CgA was elevated? Please clarify in the text.

Response: Our immunohistochemistry data with CgA clearly showed that the expression of CgA is highest at CA2 and CA3 region, in line with previous study (PMID: 12106456 and 20005907). This result also complies with our finding in patient sample where we found the CgA staining mostly in CA2, CA3 and CA4 region (E-Fig. 1C). In hippocampus, other regions exhibit sparse CgA signal. The significance of this localized presence of CgA in hippocampus is not clearly known.

7. Extended data Fig. 1K-L – Please use an additional tau aggregation marker, such as Amytracker to convincingly show tau aggregation. Generally, MC1 is a conformational tau antibody and not necessarily a marker of tau aggregation.

Response: We appreciate the reviewer's comments. Indeed, MC1 is a conformation specific Tau antibody, whose reactivity depends on both the N terminus (amino acids 7–9) and an amino acid sequence in the third microtubule binding domain of Tau (PMID: 9130141). MC1 antibody has been widely used to detect misfolded tau including both the soluble form and the insoluble form as aggregates (PMID: 24271788). In this work, our intention is to assess the level of the pathological conformation of Tau, not just the Tau aggregates, therefore using MC1 antibody is suitable. Nevertheless, as suggested by the reviewer, we have now used Amytracker and obtained comparable results to those with MC1 (please see E-Fig. 1N).

8. Extended data Fig. 2A-B – The authors show that CgA levels are higher in the CA3 region in hTau mice (Extended data Fig. 1C-D). Therefore, it is unclear why they measured brain atrophy in the dentate gyrus and CA1 regions. Please measure thickness of the CA3 for relevance to CgA expression.

Response: As suggested by the reviewer, we have now measured CA3 thickness (E-Fig. 2C).

9. Fig. 2C – It is unclear why tau phosphorylation and PSD95 levels were measured in cortical rather than hippocampal samples. Please clarify or perform Western blot analysis with hippocampal samples. This is essential for consistency and coherence with the data which indicates that CgA levels are higher in the CA3 region of the hippocampus and the EM data (Extended Figure 2J-K).

Response: As suggested by the reviewer, we have now conducted Western blot studies of hippocampal samples using pTau (S202/T205), pTau (S396/S404), Tau, PSD95 and actin antibodies. We found a significant decrease in pTau expression in CgA-KO/hTau as compared to hTau hippocampus (see new E-Fig. 2G-I). Consistent with

EM findings, we observed a significant decrease in PSD95 immunoreactivity in hTau hippocampus compared to WT and CgA-KO/hTau hippocampi (see new E-Fig. 2G&J).

10. Extended Figure 2E - PSD95 data should be shown for WT mice as the control and for consistency.

Response: We have now shown PSD95 immunoreactivity in WT hippocampus (see new E-Fig. 2F&J).

11. Extended Figure 2J-K – The synapse number should be shown for WT mice as the control and for consistency with Extended Figure 2L-M.

Response: As suggested by the reviewer, we now provided EM photograph and morphometric analysis of WT synapses (see new E-Fig. 2O-R).

12. Extended Fig. 4A-B - The rationale for using CD68 as the one microglia marker and looking at expression in the CA1 alone (rather than also looking at CA3 and the DG as done in previous figures) is not established. Please provide the data or clarify. Please also include a WT group of mice as the control.

Response: As suggested by the reviewer, we have now extended our studies to the CA3 section and included a WT group as a control. We found significant decrease in CD68 immunostaining in CgA-KO/hTau compared to hTau mice (see new Fig. 4A-D). In addition, we also immunostained with Iba1 antibody and found comparable results to those of CD68 immunolabelling (see new Fig. 4E&F). Along with this data, the reduced inflammatory cytokine profile from CgA-KO/hTau cortex strongly supports our claim that neuroinflammation is reduced when CgA is depleted.

13. On Line #177 the statement that the CgA-KO/hTau has lower levels of CD68 is the wrong interpretation given that the actual data (Extended data Fig. 4A-B) measures only area and no other aspects of microglial morphology. To claim level changes, intensity-based approaches are more appropriate.

Response: We thank the reviewer for this insightful comment. In this work, we addressed the effect of CgA depletion on Tauopathy, where neuroinflammation, a known driver of neurodegeneration, is reduced. Although we observed some morphological changes in microglia during immunostaining with Iba1 (Fig. 4E), we did not assess or compare the morphology in CgA-KO/hTau mice. Instead, our inflammatory cytokine data more strongly supports the conclusion that neuroinflammation is reduced in the hTau brain.

14. The section on metabolites is under-developed and does not cohesively fit into the rest of the story.

Response: As suggested, we have excluded metabolite data in this revised manuscript.

15. Fig. 4K – The control for the specificity of the Adra1b antibody should be included. The staining pattern for Adra1b appears to be very different between the 3 groups of mice. Please provide an explanation for this variability.

Response: The antibody we used has been cited in several peer reviewed journals (PMID: 35920096, 35537053, 34166476, 33434145, 33393489, 33118708, 32958803, 32424251, 32377606, 33088687, 31321979, 30469522, 30355157, 30277480). Moreover, the antibody has already been validated by Abcam. We found Adra1b staining strongly in the CA2 and CA3 mossy fiber region of hippocampus which is consistent in all the sections. There is some background staining which is observed in different regions are non-specific and not consistent with any cellular features.

16. Extended Fig. 8A-B - The interpretation that upregulation of Adra1 and downregulation of Adra2 contribute to elevated cAMP levels and the downstream effects is highly speculative and not directly supported by data shown in the manuscript. In fact, given that elevated levels of Adra1 signaling may lead to elevate cAMP levels and that reduced levels of Adra2 signaling may lead to reduced levels of cAMP suggests a potential zero net change in cAMP levels in hTau mice.

Response: Adra2 signaling couples to the Gi/o protein, which inhibits adenylate cyclase activity, leading to a decrease in cAMP production. Therefore, reduced levels of Adra2 would result in elevated cAMP levels. In other

words, both the upregulation of Adra1 and the downregulation of Adra2 promote an increase in cAMP levels. We acknowledge that we have not explored the Adra2 pathway in detail and, as a result, are unable to determine the relative contributions of these two pathways.

17. In Line #116-117 the author states a wrong conclusion “These results suggest that CgA promotes tau phosphorylation and aggregation.”. All data presented is from CgA KO mice, and thus, it does not directly conclude that CgA promotes the tau phosphorylation. Similar overstatements are made in line #136-137 regarding pro-neurodegeneration features of CgA. Please correct results descriptions and conclusions accordingly to match the data presented.

Response: We appreciate reviewer’s comments on this issue. Now we have included the data which depicts CgA treatment can induce formation of pathological conformation of Tau in AAV-MAPT (P301S) transduced primary neurons pre-treated with K18 fibrils (**Fig 1K-L**), which suggests that CgA promotes NFT (neurofibrillary Tau tangle) formation in neurons via direct or indirect mechanisms. As noted by the reviewer, our *in vivo* data does not involve exogenous CgA treatment. Therefore, we have modified the text accordingly.

18. On Line #135 please clarify what is meant by “irregular” synapses (i.e., shape, size, volume, structural architecture, etc.).

Response: We have now changed the text accordingly.

19. Please define what NE (norepinephrine) stands for in Line #283.

Response: We have changed the text accordingly.

20. Please provide open-access IDs for raw data files from bulk RNA-seq and if possible, attach the analyzed LogFC/p-value datasheet to the extended data. This is an important resource for the research community

Response: Open access id: [GSE274459](https://www.ncbi.nlm.nih.gov/geo/query/acc.cgi?acc=GSE274459)

We have now addressed this point and updated in supplementary tables (**Table 1 and 2**).

Minor points

1. Please add to the animal’s section the source of the mice used in the study (both the CgA-KO and hTau). Please cite the appropriate papers on how these mice were generated, characterized, and validated. Additionally, I would strongly advise authors to refer to the hTau model, or minimally introduce it, by its widely known strain name PS19.

Response: We have now addressed this point. Please see lines 430-432.

2. Please include how long (seconds/minutes) the animal was allowed to swim before being guided to the platform in the MWM.

Response: Animals are allowed to swim for a maximum of 40 sec before being guided to the platform by the experimentalist.

3. The citation on line #411 for organotypic slice cultures is not in the right citation format as the rest of the manuscript.

Response: Sorry for this inadvertent mistake. We have now provided correct citation.

4. If performed, please add a statement about data normality testing and usage of non-parametric tests (when applicable). If data normality was not performed, please justify.

Response: Addressed

5. Please include the source (company and catalog #) of antibodies used for both WB and ICC. The dilutions used are provided, but for example, it is unclear which are the CgA and total tau antibodies used and if they have been previously validated.

Response: We validated the CgA antibodies in this work (**Fig. R2**). The total Tau antibody (HT7) is a widely used Tau antibody which specifically detect human Tau, not mouse Tau.

6. Fig 1C and J should be moved to the supplements as the authors do not pursue differences in the change at the protein or transcript levels of CgA and move forward with a CgA KO model.

Response: As suggested by the reviewer, we have moved to **Fig. 1C to E-Fig. 1A**.

7. Fig 1C is cited after figures 1D-E in the results section; Extended figure 1E is cited before 1C and D. Please correct.

Response: We apologize for this oversight and thank reviewer for pointing this out. This mistake has been corrected.

8. Relative to extended figure 1H, please address the difference in molecular weight of tau in WT animals seeded with K18 relative to the other experimental conditions.

Response: Ex Fig 1H shows the OTSC transduced with AAV-P301S hTau and seeded with K18 fibril. The difference in molecular weight of tau (P301ShTau) after K18 seeding is likely due to the post translational modification, especially hyperphosphorylation of human Tau in WT brain slice cultures. We only verified phosphorylation at two epitopes (S202, S396/S404), where we observed remarkable differences. Other phosphorylation sites, as well as other PTMs such as acetylation, sulfation, glycosylation, acylation, etc., could all contribute to the prominent shift in molecular weight of tau.

9. Representative images in Extended Fig 1K and main Fig. 5E make unclear what regions are being assessed in the hippocampal OTSC, which makes the MC1 staining hard to interpret. Please include neuronal marker stain or a better image showing the CA3 region. Please address this in the text.

Response: We thank the reviewer for pointing this out. We have now included better images in Fig 1 M&N. In all the OTSC images we have annotated the CA3 and DG regions.

10. Data in Extended Fig. 3C and D should be moved to the main manuscript Fig. 3 since they add to the behavioral validity of the model.

Response: As suggested by the reviewer, we have moved **E-Fig. 3C&D to Fig. 3G&H**.

11. Relative to extended Fig. 2K-M and main Fig 3B - please show individual points for consistency with other graphs throughout the paper.

Response: As suggested by the reviewer, we have now shown **individual data points in new E-Fig. 2P&R**.

12. Relative to extended Fig. 4A and B, and main Fig. 4K the CD68 representative image in hTau mice has more background-like signal than the CgA KO. Please clarify if images were processed similarly (i.e., during acquisition and post-processing). Additionally, if there are discrepancies in the signal specificity, measurement of area in this case may not be the best endpoint and authors should consider looking at either the intensity of CD68 signal or stereological analysis of microglia number/morphology.

Response: We confirm that our images were processed in same way for each experimental set. We started with the WT and that same exposure we took the images of hTau and CgA-KO/hTau. We would like to clarify that these are Iba-1 staining rather than CD68 staining. Therefore, quantifying the percent area (Ar. Fr.) rather than the intensity is more relevant. Our previous experience suggests that the percent area strongly correlates with microglia number.

13. Please clarify why Extended Fig 5 (addressing pro-inflammatory cortex and plasma cytokines) is separate from Extended Fig. 4 and cited before Extended Fig. 4G-L, perhaps glia data should be consolidated in Extended Fig. 4 and metabolic data – now spread across Extended data Fig. 4I-L and Extended Fig. 6 - should be in a separate figure.

Response: Metabolite data is excluded from manuscript. Due to lack of space the cytokine plots are distributed in two figures.

14. Fig. 4C is mentioned before Fig. 4A and B.

Response: Sorry for this inadvertent mistake, which has been corrected now.

15. Fig. 6C is mentioned before Fig. 6A and B, please correct.

Response: Sorry for this inadvertent mistake, which has been corrected now.

Response to Reviewer #3:

Jati and colleagues present a work entitled 'Chromogranin A Deficiency Attenuates Tauopathy by Altering Epinephrine–Alpha-Adrenergic Receptor Signaling' in which they study the role of chromogranin A changes in human tauopathies and experimental models. First, they found an increased amount of chromogranin A in the autopsy-derived brain materials from patients with Alzheimer's disease or corticobasal degeneration. Such increase was also found in the brains of tau transgenic mice (PS19). They then crossed Chromogranin A knockout mice with PS19 transgenic mice. Neuropathological changes and cognitive functions were preserved in CgA-KO/PS19 mice. Transcriptomic data allowed for the identification of changes in adrenergic receptors expression. Knowing that chromogranin A plays an important role in the adrenergic system. They then studied the alpha-adrenergic receptor and epinephrine in tau transgenic mice, focusing on the inflammatory component. Their results suggest that the Alpha-adrenergic receptor/epinephrine system plays an important role in the connection between metabolic regulation and neurodegeneration.

These findings are original to the field but also related fields like metabolism.

Altogether, the work is original but the storytelling might be improved. For instance, the ex vivo work on CgA-KO brain slices in extended figure 1 panels G-L is more related to the next section in CgA-KO/PS19 mice. Regarding the data, some conclusions may be further supported by the addition of controls.

Response: We thank the reviewer for carefully reading the manuscript and providing valuable comments. We also appreciate the reviewer for recognizing our work as 'original' and 'very nice'. As suggested and outlined above in response to reviewers 1 and 2, we have performed several new experiments to address the concerns raised.

Comments:

In the field of tau research, htau mice refer to a well-described strain developed by the laboratory of Peter Davies and available from The Jackson Lab (<https://www.jax.org/strain/005491>). The strain used in the present is known as TauP301S (line PS19) (<https://www.alzforum.org/research-models/tau-p301s-line-ps19>). Please modify to avoid any confusion.

Response: We thank the reviewer for the remark. To avoid confusion, we have mentioned specifically in Introduction where we wrote about PS19 and in the entire manuscript we will use 'hTau' for PS19 "line 100-101".

In patients, there is a mix of CBD and PSP. Neuropathology indicates ARTAG suggesting glial tau positive inclusions. There is no information about CgA immunoreactivity in astrocytes. Do the authors investigate such co-localization?

Response: We appreciate the insightful comments from the reviewer regarding ARTAG. However, we believe that the potential role of CgA in astrocytes is beyond the scope of the current work and would be interesting to explore in the future.

HT7 is an antibody which recognizes human tau and not mouse tau. Is murine tau modulated in CgAKO mice? What are the levels of phosphorylation in murine tau? In the manuscript, there is no analysis of murine tau phosphorylation and expression in WT and CgAKO. It must be explored to precise what is the connection between Alpha adrenergic receptor/epinephrine and tau phosphorylation. Is it physiological or only there in tau overexpression models? Similarly, many data from CgAKO are missing. For instance in extended Figure 2,

nothing on thickness. Conversely, in inflammation studies, the four animal groups are studied. Why some groups are lacking in other experiments? Murine tau, life span...

Response: We thank the reviewer for the important point. We have now carefully evaluated our WB for the endogenous mouse Tau. As shown in Fig. 2C, there is a mild but noticeable decrease in pTau, especially pTau (S396/S404) (mouse tau is the lower band slightly below 60 kD that is present in WT mice). We understand that our experiments are majorly done in PS19 mouse model, but our key observations about increased CgA level in tauopathy mice, altered level of adrenergic receptor and catecholamine level are in synergy with the patient sample data (Fig. 1A-H, E-Fig. 5F-I, Fig. 7A-D). These results suggest that even though the data were generated from PS19 model, the mechanistic outcome is of utmost clinical importance for tauopathy related dementia. We have now included the thickness measurement for CgA-KO mice in E-Fig. 2A-C.

The use of immunoblotting for Chromogranin A quantification is not satisfactory. Many immunological assays for CgA are available since it is a well-described biomarker in oncology.

Response: We thank the reviewer for pointing this out. We have performed CgA-image analysis wherever possible. Analysis based on immunoblot and image support each other. The antibody used is being validated in Fig R1.

Surprisingly, there is no behaviour tests on fear and depression where epinephrine is a key factor. For pharmacological studies with prazosin and phenylephrine, is there any dose-response study?

Response: Our behavioral studies were done prior to our discovery of the involvement of the adrenergic pathway, which was based on transcriptome analysis using the same groups of animals that were taken down after the behavioral studies. We do, however, appreciate this comment and will further investigate fear and depression behavior in our future studies. We performed the dose response study for prazosin and phenylephrine (E-Fig. 8D&G).

The figures and extended figures are very busy and difficult to understand. In figure 1, the panel 1C of CgA mRNA expression is not explicit. The y-axis indicates normalized counts but there is no indication of mRNA (except in the figure legend).

Response: We have now labeled "normalized count (*Chga*)" in the Y axis to address the issue of CgA mRNA. We agree with the reviewer that that our figures and panels are busy. However, we tried our best to demonstrate our findings as explicitly as possible.

In figure 1D, there is no information about the area (CA or DG) and cell layers (granular cell layer, pyramidal layer, etc...) of the hippocampus where high magnification were taken. There is no scale bar in Figure 2A, and hippocampus of the CgA-KO/hTau (PS19) panel looks bigger.

Response: We thank the reviewer for the suggestion. We have now included the low magnification image from Braak6 and Braak1 sample with specified location from where the high magnification image was taken (E-Fig. 1C). The CgA expression is mostly present in CA3-CA2 region and is significantly higher in Braak6 compared to Braak1.

We have added the scale bar for Fig. 2A and replace the image for CgA-KO/hTau mice with a more representative one.

Very nice work which may be improved

Response: We very much appreciate this reviewer for recognizing the importance of our work and for his/her complimentary comment. Since we addressed most of the reviewers' comments, we believe that our revised MS has been significantly improved.

Response to Reviewer #4:

Figure R1: Series of hippocampus image for WT, hTau, CgA-KO and CgA-KO/hTau. Scale bar: 200 μm .

REVIEWER COMMENTS

We thank all reviewers for their careful reading of the revised manuscript and their additional comments to improve the quality of the MS. Below is our point-by-point response to the reviewers' comments:

Response to Reviewer #1 (Remarks to the Author):

Comment: In the revised manuscript, the authors have satisfactorily addressed most of the concerns raised in the original submission. The new data demonstrate that neuronal uptake of Tau seeds is significantly increased in the presence of CgA. Additionally, the manuscript now includes data on the colocalization of adra1B and pTau. The authors have provided validation experiments for the CgA antibody, though the corresponding figure is included only for the reviewer. The metabolic section, which was weaker in the previous version, has been removed as per the suggestion of another reviewer in consultation with the Editor, resolving some of the previously raised issues. Overall, the revisions have strengthened the manuscript, and it is now suitable for publication.

Response: We thank the reviewer for such supportive comments. We have now included the corresponding figures in supplementary figures as follows:

E-Fig 1A, B (Antibody validity)

E-Fig 7A (Depletion of Chga mRNA in CgA-KO/PS19)

E-Fig 2 (Series of hippocampus images for WT, PS19, CgA-KO and CgA-KO/PS19)

E-Fig 3 O-R (O-P; Control experiments to demonstrate that PBS injection did not induce tau spreading from ipsilateral to contralateral hippocampus. Q-R; The spreaded Tau is the endogenous Tau and not the injected K18 fibrils.)

Response to Reviewer #2 (Remarks to the Author):

The revised manuscript by Jati et al. has significantly improved following suggestions from all the Reviewers. The authors have improved the flow of the manuscript by removing the incomplete metabolite data and focusing on the tau dysregulation. They have performed an extensive number of new experiments/analyses to address concerns related to CgA antibody specificity, in vivo tau injection experiments, and directionality of the CgA-Adra1-EPI-tau axis. Additionally, they have provided improved data representations and convincing explanations throughout the rebuttal letter when new experiments were not possible. The current manuscript requires a few minor edits prior to publication (detailed below).

The points below specifically refer to the points addressed in the Rebuttal letter.

Comment: Point 2: The authors have substantially reduced concern regarding specific detection of CgA in Fig. R2. Please provide a CgA Western blot with higher exposure for Fig. 2C, similar to the robust detection observed in Fig. R2 and the other blots in Fig. 2C.

Response: We appreciate the reviewer for raising the concern. We have now provided CgA western blot with higher exposure in Fig. 2C.

Additional points

Comment: 7. We agree with the authors about the nature and description of MC1 binding, and its suitability to detect misfolded tau. However, misfolded tau is not the same as aggregated tau, and using MC1 in IHC analysis does not permit to confidently address tau aggregation. For instance, the paper cited above (PMID: 24271788) does not refer to the MC1 staining as aggregated tau, but as misfolded tau. Meanwhile, the current manuscript extrapolates the MC1+ to tau aggregation in multiple places such as lines #138,159,274,276,298-306,312,339,342,349,473,and 658. Please revise the statements in the text to appropriately address the misfolded species of tau that is labeled with MC1.

Furthermore, the suggestion to include validated markers (e.g., Amytracker), which detect features of tau aggregates (i.e., amyloid structure) was to allow the authors to confidently establish the effects on tau aggregation and not just misfolding. However, the image provided for the Amytracker staining in E-Fig. 1N is not convincing and looks like non-specific signal in absence of appropriate controls (i.e., no dye, dilution curve, more sample size, etc.). We appreciate the effort of the authors to provide this piece of data, but in its current state, it should be removed from the manuscript. The authors should revise the language in the manuscript to “misfolded tau” as discussed above or provide more convincing data for the formation of true tau aggregates that form amyloid structure.

Response: We thank the reviewer for such constructive and insightful comments. We removed Amytracker staining figure from the amended manuscript. We have now replaced the term “aggregated tau” with “misfolded tau”/“misfolded tau species” throughout the manuscript.

Comment: 13. This was not addressed and the previous phrase (now line 213) still incorrectly states decrease in levels of CD68 while only area was measured. Please correct.

Response: We thank the reviewer for bringing up this concern. Yes, we have now corrected those regions mentioning about ‘area fractions’ we measured. Lines 217-220.

Comment: 20. The point was addressed. However, it is highly recommended that, upon publication, the tables are provided as searchable Excel files instead of tables in PDF files as it is currently presented.

Response: We have submitted the excel files. But the submission portal generated the pdf.

Minor points

Comment: 1. The point was addressed for the CgA KO mice. However, as pointed also by Reviewer #3, the authors should modify the hTau nomenclature to PS19 because that

is the widely known name of the strain and referring to it as hTau gives the wrong impression that these mice express non-mutated human tau.

Response: We have now changed the nomenclature from 'hTau' to 'PS19' throughout the manuscript and figures.

Comment: 2. Please include this information in the methods section of the MWM.

Response: We have now included this information in line 548-550.

Comment: 4. This was not addressed nor added to the statistical methods section.

Response: We thank the reviewer for bringing up this concern. We have now included this in Statistics section (line 704-705).

Response to Reviewer #3 (Remarks to the Author):

Comment: The authors' answer is satisfactory. However, since no immunological tests other than immunoblot were used, it would be desirable to visualise the entire blot in order to identify the fragments derived from chromogranin A.

Fig. R1. Entire Western blot for Chromogranin A (CgA). A represents entire blot for Fig. 1A; B represents entire blot for Fig. 1B ; C represents entire blot for Fig. 1G; D represents entire blot for Fig. 1I.

Response: We have now included Fig. R1 here to address this concern. We sincerely hope that this will satisfy the reviewer and the editor. Nevertheless, we can submit the entire blot for all the immunoblot figures if requested.

Response to Reviewer #4 (Remarks to the Author):

Response: There were no concerns to address.